



# Inferring sediment-discharge event types in an alpine catchment from sub-daily time series

Amalie Skålevåg[1], Oliver Korup[1], and Axel Bronstert[1]

[1]Institute of Environmental Science and Geography, University of Potsdam, Germany

*Correspondence to:* Amalie Skålevåg (skalevag2@uni-potsdam.de)

**Abstract.** Fluvial sediment dynamics in mountain rivers are changing rapidly in a degrading cryosphere, raising the potential for erosive rainfall and runoff, and detrimental effects on downstream areas. Hence, we need to understand better what char-
acterises and drives episodic pulses of water and suspended solids in rivers. Here, we infer different types of such sediment-discharge events from 959 automatically detected events based on 16 metrics derived from 15-min time series of streamflow and suspended sediment concentrations from the Vent-Rofental in the High Ötztal Alps, Austria. We use principal component analysis to extract uncorrelated event characteristics and cluster event types with a Gaussian mixture model. We interpret thus inferred event types with catchment metrics describing antecedent conditions, hydrometeorological forcing, and catchment
freezethaw state and snowcover. We find event magnitude, hysteresis, and event shape complexity to be the main factors characterising the overall event regime. The most important characteristics distinguishing the event types are suspended sediment and streamflow magnitude, and event shape complexity. Sediment-discharge hysteresis is less relevant for discerning event types. We derive four event types that we attribute to (1) compound rainfall-melt extremes, (2) glacier and seasonal snow melt, (3) freezethaw-modulated snow-melt and precipitation events, and (4) late season glacier melt. Higher magnitude glacier
and snow melt events were the most frequent and contributed some 40 % to annual suspended sediment yield on average; compound rainfall-melt extremes were rarest, but contributed the second highest proportion (26 %). Our approach represents a reproducible method for objectively estimating the variety of event-scale suspended sediment dynamics in mountain rivers, which can provide insights into the contribution of different drivers to annual sediment yields in current and future regimes. Our findings highlight the importance of both meltwater and rainfall-runoff as drivers of high magnitude suspended sediment
fluxes in mountain rivers.

## 1 Introduction

High mountain areas have been warming at a faster rate than the global average (Hock et al., 2019), drastically changing the mountain cryosphere in terms of accelerated glacier mass loss (Hugonnet et al., 2021; Huss and Hock, 2018), permafrost degradation (Smith et al., 2022; Biskaborn et al., 2019), and snowpack reduction (Hanzer et al., 2018; Beniston et al., 2018;
Carrer et al., 2023). This ongoing cryospheric decay combines with altered precipitation and weather patterns, and change sediment dynamics and loads in mountain regions (Zhang et al., 2022). Elevated sediment loads can have detrimental effects





on water quality, hydro-power production, and aquatic habitats and ecosystems in downstream reaches especially (Adler et al., 2022; Huss et al., 2017; Scheurer et al., 2009).

Current changes in fluvial suspended sediment transport in mountain rivers are mainly affected by sub- and proglacial sources in the wake of deglaciation and reworking of freshly exposed deposits (Schmidt et al., 2022; Zhang et al., 2022; Ballantyne, 2002; Delaney and Adhikari, 2020; Delaney et al., 2018a; Hinderer et al., 2013). Episodic sediment pulses, often caused by rainstorms, can contribute substantial fractions to annual sediment yields (Vercruysse et al., 2017; Gonzalez-Hidalgo et al., 2013; Schmidt et al., 2022). Observed and projected increases in extreme precipitation (Madsen et al., 2014; Vergara-Temprado et al., 2021; Fowler et al., 2021) make it seem likely that such sediment fluxes may become more dominant, leading

to more flashy sediment-transport regimes (Zhang et al., 2022). Paraglacial environments in particular host large amounts of unconsolidated sediment that can remain available for mobilisation during extreme rainfall events long after glaciers have melted (Zhang et al., 2022; Huss et al., 2017). Thus, any gradually decaying sub- and proglacial sediment discharge may be supplemented by rainfall-driven reworking of sediment (Zhang et al., 2022). Consequently, we need to understand better the current drivers of episodic sediment fluxes in high mountain areas, and to which extent hydrometeorological forcing, sediment

availability and reworking, will affect our projections of future rates and regimes sediment-transport.

    Detailed event-based analysis of suspended sediment dynamics in mountain rivers can identify important antecedent conditions and drivers (Vercruysse et al., 2017). However, the complex and nonlinear nature of suspended sediment transport in mountain rivers poses a challenge for such analyses (Vercruysse et al., 2017; Bracken et al., 2015): the complexity arises from (1) multiple hydrological drivers of sediment transport, e.g. rainfall, snow-melt, and glacier melt (Costa et al., 2018; Orwin

et al., 2010); (2) catchment conditions and processes regulating sediment production and availability, e.g. snow and vegetation cover, freeze-thaw cycles and erosion, lithology and glacial history (Schmidt et al., 2022; Rengers et al., 2020); and (3) hillslope and channel geomorphology that influences erosion potential and sediment connectivity (Bracken et al., 2015). Hence, suspended sediment concentrations in mountain rivers are highly variable (Schmidt et al., 2022; Lalk et al., 2014; Hinderer et al., 2013; Delaney et al., 2018b). By systematically detecting and grouping episodic suspended sediment fluxes, which we

term "sediment-discharge events", we might derive a catchment-specific event typology, in which each type shares similar and dominant hydrometeorological drivers and geomorphic catchment conditions.

    While studies of event-scale suspended sediment dynamics are common, only a handful of studies have tried to identify specific event types and their conditions and drivers. Most of these studies focused on classifying patterns of sediment-discharge hysteresis (e.g. Hamshaw et al., 2018; Tsyplenkov et al., 2020; Haddadchi and Hicks, 2021), and attributed these classes

to drivers such as hydrometeorological forcing (e.g. rainfall intensity and amount); antecedent catchment conditions (e.g. soil moisture, precipitation); land cover; sediment exhaustion; or contributions from multiple sediment sources. Sediment-discharge hysteresis is a well-established concept in fluvial sediment transport research dating back to 1953 (Malutta et al., 2020). Despite its popularity in classifying event-scale sediment discharge dynamics, the interpretation of hysteresis remains contextual (Vercruysse et al., 2017) and often without direct indication of its cause (Tsyplenkov et al., 2020), especially where

the same cause is attributed to different types of hysteresis (Tab. 1 in Malutta et al., 2020).





A more general approach can be taken by clustering sediment-discharge events based on characteristics derived from their hydro- and sedigraphs. Leggat et al. (2015) and Orwin and Smart (2004) used a combined classification, separately clustering sedigraph shapes and magnitudes to identify event types and the associated dominant meteorological conditions and drivers. Javed et al. (2021) grouped events based solely on hydro- and sedigraph shapes, by normalising magnitude and standardising event lengths, and subsequently clustering with K-medoids and dynamic time warping. Mather and Johnson (2015) clustered event shapes based on sediment rating curve parameters. The advantage of using clustering for inferring event typologies is that it does not require any previous knowledge about event types (Tarasova et al., 2019). Thus, clustering is a suitable first approach for inferring sediment-discharge event types in high mountain catchments on the basis of similar water and sediment discharge characteristics.

Here, we use a clustering approach to derive a sediment-discharge event typology for the high-alpine, glaciated basin Vent-Rofental, Ötztal Alps, Austria. With its long monitoring history, the catchment has a wealth of hydrological, meteorological, and glaciological data (Strasser et al., 2018). Key to our data-driven approach are continuous, high resolution records of suspended sediment concentration since 2006. Recent reconstructions and projections of annual suspended sediment yield for Vent-Rofental basin suggest that the basin has entered a phase of declining glacial influence on sediment transport (Schmidt et al., 2023b, a), making it an ideal study area to examine. For our clustering approach, we assume that each event belongs to a certain type that shares a set of similar sediment-discharge characteristics. We identify and condense these characteristics with a principal component analysis on 16 event metrics describing event magnitude, hysteresis, shape, and effects of preceding events. We cluster events based on these characteristics with a Gaussian mixture model, and use hydrometeorological data to interpret each cluster as an event type. Our aim is to understand the catchment state, antecedent conditions and hydrometeorological drivers that determine event-scale suspended sediment dynamics in the upper Ötztal, with the following research questions:

- what are the key sediment-discharge characteristics that differentiate the event types?

- how well can we identify shared hydrometeorological drivers of events of the same type?

- are the event types associated with diagnostic antecedent conditions (e.g. dry vs. wet, cool vs. warm)?

- what is the contribution of each event type to the annual suspended sediment yield?

## 2 Study area and data

Rofental is a valley located upstream of the village of Vent in the Ötztal Alps, Austria (Fig. 1. The valley has been the site of several hydrometeorological and glaciological studies in the past 150 years, and has a unique time series of long-term observations (see Strasser et al., 2018, for detailed description). The Vent-Rofental hydrological basin has 98 km$^2$ and an elevation range of 1891-3772 m a.s.l. The main river is the Rofenache, a tributary of the Venter Ache, Ötztaler Ache, and the Inn. The current hydrological regime is dominated by snow and ice melt, which peaks in July and August and is lowest during winter (Strasser et al., 2018; Schmidt et al., 2022). The seasonality of suspended sediment has a pattern similar to that





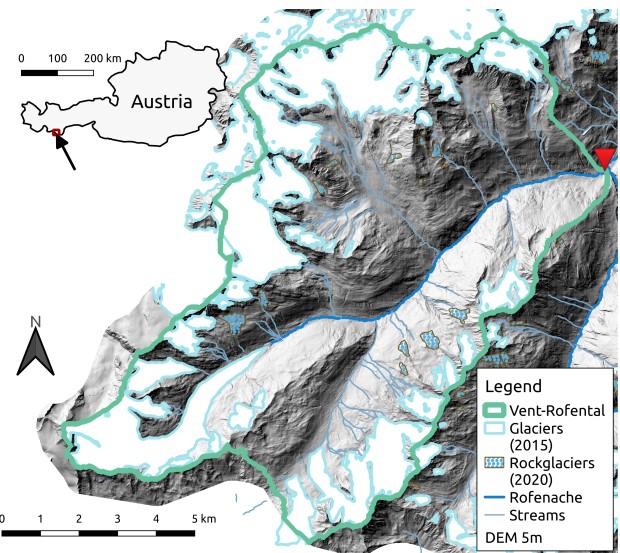

**Figure 1.** Map of study area showing topography (Land Tirol, 2021), river network (OpenStreetMap, 2022), glaciers (Buckel and Otto, 2018) and active rock glaciers (Wagner et al., 2020). The location of the Vent-Rofental river gauge is denoted with a red triangle, and the accompanying catchment boundary in green.

of streamflow, but with a longer low-flux winter period (Schmidt et al., 2022). Glaciers cover about a quarter of the catchment (28 % in 2015, Schmidt et al., 2022; Buckel and Otto, 2018). However, their size is rapidly decreasing, and they will likely

disappear by the end of the 21st century (Hanzer et al., 2018).

The bedrock consists of various types of gneiss-mica schists and schistose gneisses (Moser, 2016; Kreuss, 2018). The Quaternary geology is dominated by Holocene and some Pleistocene moraines (Moser, 2016; Kreuss, 2018). The catchment also has many talus slopes sitting between the moraines and steep bedrock slopes (Moser, 2016; Kreuss, 2018). Apart from a few small lakes and proglacial outwash plains, the river network lacks significant sediment storage.

The Vent-Rofental river gauge (1891 m a.s.l., 46.85691°N, 10.91093°E) has been operated continuously by the Hydrographic Service of Tyrol (HD-Tirol) since 1967, and suspended sediment concentrations (SSC) have been monitored since 2006 with two optical infrared turbidity sensor (Solitax sensors by Hach) and manual SSC sampling (see Lalk et al., 2014, for details). The continuous SSC monitoring at Vent-Rofental was established as part of a nation-wide strategy by the Austrian Hydrographic Service in order to monitor and analyse changes in riverine suspended sediment resulting from deglaciation,

permafrost thawing, land use changes and river regulation (Habersack et al., 2008; Lalk et al., 2014). SSC samples to calibrate turbidity measurements were collected manually close to the turbidity sensors frequently, and, when possible, during high flow events (Lalk et al., 2014). Turbidity measurements at the Vent-Rofental gauge are paused in winter (October-April) to prevent damage to the equipment. However, the sensors are installed before the spring rise in SSC, and the sediment transport during the unmonitored period can be considered negligible.



**Table 1.** Description of datasets and variables used in this study. Temporal extents refer to available data in the study period (2006-2021).

| Dataset | Temporal extent | Description | Source |
|---------|-----------------|-------------|--------|
| SSC | 2006-2021 | 15-min turbidity-based SSC observations at Vent-Rofental gauge | HD-Tirol |
| Q | 2006-2021 | 15-min streamflow observations at Vent-Rofental gauge | HD-Tirol |
| SPARTACUS | 2006-2021 | 1-km daily maximum and minimum air temperature, interpolated from point observations | GeoSphere Austria |
| INCA | 2006-2021 | 1-km hourly precipitation, station-adjusted weather radar observations | GeoSphere Austria |
| APOLIS | 2006-2021 | 100-m daily global radiation | GeoSphere Austria |
| SNOWGRID | 2006-2021 | 1-km daily modelled SWE and snow depth | GeoSphere Austria |
| WINFORE | 2006-2021 | 1-km daily $SPEI_{30}$ | GeoSphere Austria |
| MODIS-SC | 2006-2018 | 250-m daily observed snowcover maps | Matiu et al. (2019) |

A number of gridded products of hydrometeorological variables are available for the Rofental (Tab. 1), mostly from the Austrian Weather Service *GeoSphere Austria*. We used these data to calculate various metrics of catchment conditions and processes during and leading up to events (see Sec. 3.4).

SPARTACUS provides daily maximum and minimum temperature fields at 1-km resolution based on interpolated station data. Time series from 150 stations in Austria are interpolated with a method that combines a macroclimatic background field

with a mesoclimatic residual field (see Hiebl and Frei, 2016, for details).

WINFORE provides daily reference (potential) evapotranspiration fields at 1-km resolution using a recalibrated Hargreaves method forced with SPARTACUS minimum and maximum temperature fields (see Haslinger and Bartsch, 2016, for details). The daily reference potential evapotranspiration and SPARTACUS interpolated daily precipitation fields (Hiebl and Frei, 2018) are then used to calculate 30-day standardised precipitation and evapotranspiration index ($SPEI_{30}$) fields at daily resolution.

The INCA system produces analysis and nowcasting fields for various meteorological variables. The precipitation analysis incorporates rain gauge measurements, radar data, and elevation effects. The uncorrected radar field is partially corrected to produce a climatologically adjusted radar field, which is subsequently re-scaled based on the comparison between station observations and the radar field at the station location (see Haiden et al., 2011, for details).

Daily global radiation fields are obtained from APOLIS, a 100-m gridded dataset produced by calculating direct and diffuse

solar radiation with the parametric solar radiation model STRAHLGRID (Olefs and Schöner, 2012).

SNOWGRID-CL, a daily and longer term version of the physically based and spatially distributed snow model SNOWGRID, uses an extended degree-day scheme to calculate snow ablation and a two-layer scheme to account for snow sublimation, settling and refreezing of the snow cover (Olefs et al., 2020). The model is forced with daily WINFORE potential evapotranspiration (Haslinger and Bartsch, 2016), SPARTACUS temperature (Hiebl and Frei, 2016) and precipitation (Hiebl and Frei,

2018) fields, producing daily snow water equivalent (SWE) and snow depth estimations at 1-km resolution. To complement the





SNOWGRID data, we also included daily 250-m snowcover maps derived from MODIS imagery using snow cover and cloud removal algorithms tailored to the European Alps (see Matiu et al., 2019, for details).

## 3 Methods

Our approach for identifying sediment-discharge event types is divided into three steps: (1) the detection and characterisation
of events (Sec. 3.1; 3.2); (2) grouping similar events via clustering (Sec. 3.3); and (3) evaluation and interpretation of these clusters as event types (Sec. 3.4).

### 3.1 Event detection

There is no commonly used definition of what constitutes a sediment-discharge event, i.e. an episodic suspended sediment flux measured at the catchment outlet. Studies use various terms, e.g. flood events (Pagano et al., 2019; Francke et al., 2008),
hydrologic(al) events (Tsyplenkov et al., 2020; Williams, 1989), or storm events (Javed et al., 2021; Hamshaw et al., 2018), as events are generally separated based on streamflow either by hydrological day (e.g. Antoniazza et al., 2022; Leggat et al., 2015), with hydrograph separation (e.g. Haddadchi and Hicks, 2020; Tsyplenkov et al., 2020), or with a semi-automated procedure (e.g. Hamshaw et al., 2018). In this study we use the term sediment-discharge event, which we define as a marked increase in streamflow accompanied by a large pulse of suspended sediment measured at the catchment outlet (i.e. gauge).
Continuous 15-min time series of streamflow $Q_t$ and suspended sediment concentration $SSC_t$ were used for the event detection. The start $t_0$ and end $t_1$ of an event $i$ were derived from $Q_t$ and then subsequently filtered based on event $SSC_t$ with the following procedure (Fig. 2a):

1. The demarcation of hydrological events followed the method after Tsyplenkov et al. (2020) based on the local minimum hydrograph separation method (Sloto and Crouse, 1996) using the *loadflux* R-package (Tsyplenkov, 2022). This method
essentially splits the entire streamflow record into events at local minima identified in a centered 21-hour search window $w$.

2. We removed hydrological events with no or only partial SSC measurements.

3. Only events where peak SSC exceeded a fixed threshold $\theta_{SSC,peak}$ were kept. To focus on large events we set the threshold at the 90th percentile $P_{90}$ of $SSC_t$, which for our 16-year study period (2006-2021) was 1196.5 $mg\ l^{-1}$.

$$\theta_{SSC,peak} = P_{90}(SSC_t) \tag{1}$$

This event detection procedure ensures that events of varying duration can be detected (Fig. 3), and that the detected events are extreme enough (in terms of SSC) to be of interest. With the use of threshold to filter out events, the approach is similar to a peak-over-threshold (POT) approach, except that the boundaries of our events are determined by the event hydrograph.





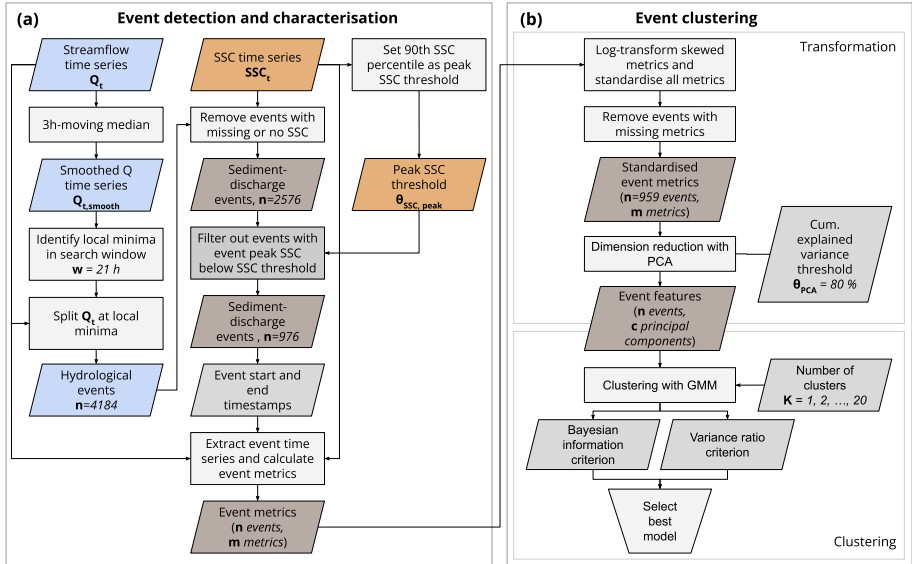

**Figure 2.** Overview of event detection and characterisation (a), and event clustering (b). Events are demarcated at local streamflow minima, then filtered by suspended sediment concentration (SSC) magnitude. Each event is then characterised with sediment-discharge event metrics (see Tab. 2). After transforming and standardising the metrics, the dimensions are reduced with principal component analysis (PCA), and clusters identified with a Gaussian mixture model (GMM). The optimal number of clusters are chosen using two objective criteria. The values specific to this study are highlighted in *italic*. Input variables and thresholds are highlighted in **bold**.

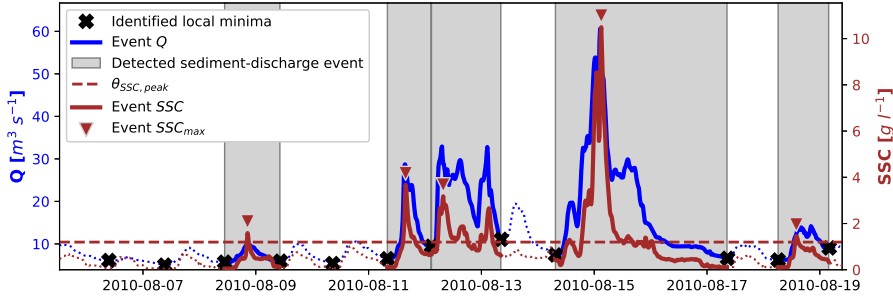

**Figure 3.** Example of event detection procedure. SSC-Q time series is split at local streamflow minima and events with peak SSC below threshold $\theta_{SSC,peak}$ are discarded.

## 3.2 Characterisation of events

In order to identify groups of similar sediment-discharge events, we need metrics to characterise them first. A number of metrics and indices have been developed to characterise event-scale (suspended) sediment dynamics. We select 14 metrics from the literature and introduce two new metrics (Tab. 2), broadly divided into four categories relating to time and seasonality, event magnitude, intra-event dynamics, and inter-event effects on sediment dynamics.





**Table 2.** Sediment-discharge event metrics used to characterise events.

| Category | Metric | Description | Unit |
|---|---|---|---|
| Time and seasonality | $\Delta t$ | Duration of the event | $h$ |
| | $DOY$ | Seasonality expressed as day of the year | - |
| Magnitude[†] | $SSC_{max}$ | Maximum (peak) SSC | $mg\ l^{-1}$ |
| | $SSC_{mean}$ | Average SSC of entire event | $mg\ l^{-1}$ |
| | $SSC_{mean,w}$ | Streamflow weighted average of SSC, i.e. $SSY$ divided by $Q_{total}$ | $mg\ l^{-1}$ |
| | $SSY$ | Suspended sediment yield | $t$ |
| | $Q_{max}$ | Maximum (peak) streamflow | $m^3\ s^{-1}$ |
| | $Q_{mean}$ | Average event streamflow | $m^3\ s^{-1}$ |
| | $Q_{total}$ | Total event streamflow volume | $m^3$ |
| Intra-event dynamics | $SHI$ | Simple hysteresis index, standardised between -1 and 1 | - |
| | $AHI$ | Aich's hysteresis index | - |
| | $\phi_{peak}$ | Peak phase difference, indicates the hysteresis direction | - |
| | $SSY_{ratio}$ | Log-ratio of SSY in the falling and rising limb of the hydrograph | - |
| | $SQPR$ | Log-ratio of number of SSC peaks to number of streamflow peaks | - |
| Inter-event effects | $IEI$ | Log-ratio of SSC peak of last event to time since last event | - |
| | $Q_{peak,ratio}$ | Log-ratio of peak streamflow of current to last event | - |

† log-transformed during cluster analysis preprocessing

Metrics of event duration and magnitude are commonly used in studies of event-scale sediment dynamics, such as the
average and peak SSC and streamflow during an event (Tab. 2). The suspended sediment yield $SSY$ (also: suspended sediment
load) is the total mass of suspended sediment passing the gauge, calculated by integrating the product of $SSC_t$ and $Q_t$ for each
timestep $dt$ between the start $t_0$ and end $t_1$ of the event:

$$SSY = \int_{t0}^{t1} SSC_t Q_t dt \tag{2}$$

For comparability with other study areas, we sometimes report specific SSY (sSSY) in $t\ km^{-2}$, which is SSY divided by
catchment area. Similarly, when we report on annual (s)SSY $t_0$ will be the start and $t_1$ the end of the year.

Hysteresis results from differences in SSC in the rising and falling limbs of the hydrograph and is commonly used to describe
event-scale suspended sediment dynamics (Malutta et al., 2020; Vercruysse et al., 2017). A number of hysteresis indices (HI)
have been developed to quantify and classify hysteresis patterns objectively (e.g. Aich et al., 2014; Langlois et al., 2005;
Tsyplenkov et al., 2020; Lawler et al., 2006). Here, we use Aich's HI (AHI) (Aich et al., 2014) and the simple HI (SHI)
(Tsyplenkov et al., 2020): both are normalised or standardised for better comparison. More positive values of AHI and SHI





indicate stronger clockwise hysteresis, while more negative values indicate more pronounced anti-clockwise hysteresis. Both metrics were computed with the *loadflux* R-package (Tsyplenkov, 2022).

Three further metrics were used to characterise intra-event dynamics, i.e. the peak phase difference, the SSC to streamflow peak ratio, and the falling and rising limb SSY ratio. The peak phase difference $\phi_{peak}$ (Haddadchi and Hicks, 2021) is a dimensionless measure of the time difference between the peaks of streamflow $t_{Q_{max}}$ and SSC $t_{SSC_{max}}$,

$$\phi_{peak} = \frac{t_{Q_{max}} - t_{SSC_{max}}}{\Delta t} \tag{3}$$

and indicates to what degree the SSC peak leads (positive $\phi_{peak}$) or lags (negative $\phi_{peak}$) the Q peak. Thus, it also indicates the direction of the hysteresis pattern.

To indicate whether the exported sediment tends to be delivered before or after the streamflow peak we include a modified version of the $SSY_{ratio}$ (Haddadchi and Hicks, 2020), which is the log-ratio of $SSY$ in the rising and falling limb of the hydrograph,

$$SSY_{ratio} = \log \left( \frac{SSY_{falling}}{SSY_{rising}} \right) \tag{4}$$

indicating whether the bulk of the sediment is delivered in the rising ($SSY_{ratio} < 0$), or falling limb of the hydrograph ($SSY_{ratio} > 0$).

The SSC to streamflow peak ratio (SQPR), defined as

$$SQPR = \log \left( \frac{n_{SSCpeaks}}{n_{Qpeaks}} \right) \tag{5}$$

indicates whether more SSC peaks ($n_{SSCpeaks}$) or streamflow peaks ($n_{Qpeaks}$) occur during an event. Negative values indicate more streamflow peaks, while positive values indicate more SSC peaks. If the SQPR is zero, the event had the same number of streamflow and SSC peaks. The peaks were identified automatically (Virtanen et al., 2020, `scipy.signal.find_peaks`) based on two criteria: the distance between peaks and the prominence of the peak. The peak prominence "measures how much a peak stands out from the surrounding baseline of the signal and is defined as the vertical distance between the peak and its lowest contour line" (Virtanen et al., 2020, `scipy.signal.peak_prominences`). To calculate the SQPR we set a minimal peak prominence of $500 \,\mathrm{mg\,l^{-1}}$ for SSC peaks, and $2 \,\mathrm{m^3\,s^{-1}}$ for streamflow, and for both a minimum distance of 1 hour between peaks.

Inter-event effects such as sediment accumulation or exhaustion on suspended sediment transport are less often considered in studies of event-scale sediment dynamics than hysteresis. Yet the amount of sediment available for transport for a certain event is likely to be affected by the time since and the magnitude of the last event. A large magnitude event may exhaust the sediment stores in the catchment, so that less sediment is available for transport during the next event. Conversely, if a long time has passed since the last event, sediment stores may have been replenished. We attempt to account for these inter-event effects with a new metric, the inter-event index (IEI), defined as the ratio of the SSC peak of the previous event $SSC_{max}^{i-1}$ in $\mathrm{mg\,l^{-1}}$ and the time between the end of the last event $t_1^{i-1}$ and the start of the current event $t_0^i$ in hours:

$$IEI = \log \left( \frac{SSC_{max}^{i-1}}{t_0^i - t_1^{i-1}} \right) \tag{6}$$





Another metric of inter-event effects is the previous (peak) flow ratio $Q_{peak,ratio}$ introduced by Haddadchi and Hicks (2020):

$$Q_{peak,ratio} = \log\left(\frac{Q_{max}^{i-1}}{Q_{max}^i}\right) \tag{7}$$

We modified this metric by using the log-ratio, rather than the ratio, of streamflow peak of the last event $Q_{max}^{i-1}$ and the current event $Q_{max}^i$.

### 3.3 Event clustering

Our approach for inferring the event types is based on the assumption that each event type shares a set of defining antecedent conditions, hydrometeorological drivers, and catchment states, which we capture sufficiently by our choice of event metrics. Clustering, a type of unsupervised machine learning analysis where data points are grouped into clusters based on their similarity (Murphy, 2012), is suited for our purposes as it does not require predefined class criteria. By clustering based on event metrics, we hope to find groups of events, i.e. event types, that are the expression of a certain set of catchment conditions and hydrometeorological drivers. We employed a two-step approach (Fig. 2b) consisting of a transformation step and a clustering step.

In the transformation step, we applied a principal component analysis (PCA) to the sediment-discharge event metrics. PCA transforms correlated metric variables into the same number of uncorrelated variables called principal components (PCs) (Kim and Kim, 2012). By selecting the top PCs ranked by their explained variance, we end up with a re-projected dataset, where the variables, i.e. PCs, contain most of the variance from the original variables.

After the event detection and characterisation, we ended up with a $n \times m$ data matrix of $n$ sediment-discharge events and $m$ event metrics. Some of these metrics describe similar aspects of the same event property, e.g. event magnitude or hysteresis. By transforming the dataset with PCA we achieve two objectives. Firstly, we obtain $c$ principal components (PCs) that can be interpreted as uncorrelated event characteristics. Secondly, we reduce the dimensions of our dataset by selecting fewer PCs than metrics. Preprocessing the event metrics was necessary before applying the PCA. Event metrics describing magnitude were natural-log-transformed (Tab. 2) as the distributions of these metrics were highly skewed. Next, all metrics where standardised by subtracting the mean and dividing by the standard deviation. This step is needed before performing a PCA, since the principal components can be "misled" by directions in which the variance is high merely because of the measurement scale (Murphy, 2012). Finally, the metrics were re-projected in a lower dimension space with PCA, by keeping only those ranked principal components that together accounted for more than 80 % of the total explained variance. This left a $n \times c$ data matrix $\boldsymbol{X} = (x_{ij})_{1 \leq i \leq n,\ 1 \leq j \leq c}$, where $c < m$.

In the clustering step, the event types were inferred with a Gaussian mixture model (GMM). A GMM is a mixture of $K$ multivariate Gaussians $\mathcal{N}$ with mean vectors $\boldsymbol{\mu}_k$, denoting the center of the cluster in all $c$ dimensions, and covariance matrices $\boldsymbol{\Sigma}_k$ denoting the shape of the cluster, and mixing weights $\pi_k$, such that

$$p(\boldsymbol{x}_i \mid \lambda) = \sum_{k=1}^{K} \pi_k \mathcal{N}(\boldsymbol{x}_i \mid \boldsymbol{\mu}_k, \boldsymbol{\Sigma}_k) \tag{8}$$





The mixture weights satisfy the constraint that $\sum_{k=1}^{K} \pi_k = 1$. The parameterisation $\lambda$ of the GMM is given by:

$$\lambda = \{\pi_k, \boldsymbol{\mu}_k, \boldsymbol{\Sigma}_k\} \quad k = 1, ..., K \tag{9}$$

The GMM is a type of soft clustering that estimates the probability of each event $\boldsymbol{x}_i = \boldsymbol{X}_{i\star}$ belonging to each cluster $k$. Each event is assigned to the cluster to which is has the highest likelihood of belonging.

The GMM can be fitted with different covariance types for $\boldsymbol{\Sigma}_k$. We tested three different options: a full covariance matrix
with dimensions $k \times c \times c$, a diagonal covariance matrix with dimensions $k \times c$, and a spherical covariance matrix with dimensions $k$. The full covariance type is the most flexible model by admitting independent Gaussians for each cluster, while the spherical is the most restrictive allowing only one variance value for each cluster.

In order to determine the optimal number of clusters $K$ and covariance type we used two objective criteria, the Bayesian Information Criterion (BIC) (Schwarz, 1978), which penalises higher number of parameters needed to describe more clusters,
and the Variance Ratio Criterion (VRC) (Caliñski and Harabasz, 1974), where a higher value relates to a model with better defined clusters. The selection of the optimal cluster model was based on the objective criteria of the lowest BIC and highest VRC. However, depending on the agreement between the BIC and VRC scores, the final choice may require expert judgement, e.g. by use of the elbow method.

### 3.4   Interpretation of event clusters with catchment metrics

For the interpretation of the inferred event clusters, we selected a number of catchment metrics describing antecedent conditions and hydrometeorological forcing; all these may be relevant for suspended sediment dynamics in mountain rivers (Tab. 3; Tab. 1).

In order to measure differences between event types consistently, in terms of both sediment-discharge characteristics and hydrometeorological catchment conditions, we standardised all event and catchment metrics such that

$$z_i = \frac{x_i - \overline{x}}{s} \tag{10}$$

where $x$ is the metric, $\overline{x}$ its mean and $s$ its standard deviation across all events. Such standardised $z$-scores are useful to compare groups of events (Javed et al., 2021).

## 4   Results

### 4.1   General characteristics of detected sediment-discharge events

Our detection routine identified 976 sediment-discharge events between 2006 and 2021. On average, the annual event frequency is 60 events per year. While events occurred throughout the monitored suspended sediment season from May to October, most happened from mid-June to early September (Fig. 4a), when daily suspended sediment export was highest following the snow- and glacier melt season. During this period, the peak suspended sediment concentration threshold $\theta_{SSC,peak}$ of 1196.5 mg l$^{-1}$



**Table 3.** Catchment metrics describing catchment conditions and hydrometeorological drivers leading up to and during sediment-discharge events.

| Category | Metric | Description | Unit | Data |
|---|---|---|---|---|
| Water | $NAPI_{14}$ | Normalised antecedent precipitation index (Heggen, 2001), indicating the moisture conditions in the catchment over the last 14 days leading up to the event. | - | INCA |
| | $SPEI_{30}$ | Standardised Precipitation Evapotranspiration Index (SPEI) of the 30 days leading up to event | - | WINFORE |
| | $I_{max}$ | Maximum precipitation intensity, maximum of maximum grid cell in each event time step. | $mm\,h^{-1}$ | INCA |
| | $P_{total}$ | Total catchment average precipitation | $mm$ | INCA |
| | $SM$ | Snowmelt, as estimated from change in mean catchment snow water equivalent (SWE)[†], from first to last day of event. | $mm$ | SNOWGRID |
| | $SA$ | Snow accumulation, as estimated from change in mean catchment SWE[†], from first to last day of event. | $mm$ | SNOWGRID |
| Energy | $FCF$ | Frost change factor, average area affected by diurnal freezethaw during event, i.e. daily maximum air temperature above $0\,°C$ and daily minimum air temperature below $0\,°C$ | - | SPARTACUS |
| | $ATI_5$ | Antecedent thawing index , the thawing index (Frauenfeld et al., 2007) in the 5 days leading up to the event. | degree-days | SPARTACUS |
| | $AFI_5$ | Antecedent freezing index, the freezing index (Frauenfeld et al., 2007) in the 5 days leading up to the event. | degree-days | SPARTACUS |
| | $AGR_5$ | Antecedent global radiation, average global radiation in the 5 days leading up to event | $kWh\,m^{-2}$ | APOLIS |
| | $T_{max}$ | Maximum daily maximum temperature of event | $°C$ | SPARTACUS |
| | $GR_{event}$ | Average global solar radiation during the day(s) of the event | $kWh\,m^{-2}$ | APOLIS |
| Catchment state | $fSCA$ | Fraction of catchment area with snowcover | - | SNOWGRID[‡], MODIS-SC |
| | $ACDA$ | Actively contributing drainage area (ACDA Li et al., 2021), fraction of total catchment area with above $0°C$ temperatures | - | SPARTACUS |

† Excluding glaciated catchment area (Buckel and Otto, 2018)

‡ Snowcover in SNOWGRID defined as grid cells with snow depth > 0.01 m

was exceeded frequently. The largest events occurred towards the end of the glacier melt season, i.e. between mid-July and

August.





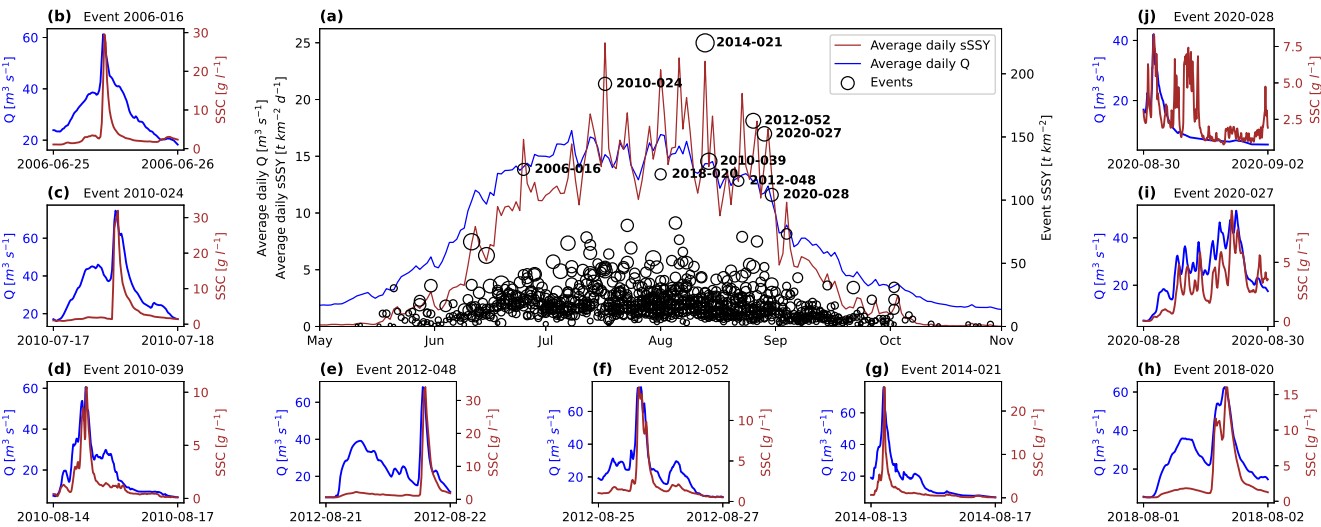

**Figure 4.** Seasonality and magnitude of detected sediment-discharge events (a), including the average annual cycle of daily streamflow (Q) and specific suspended sediment yield (sSSY) for the study period (2006-2021), together with the timing and sSSY magnitude of events, whose size is proportional to event total streamflow volume. The nine events with (s)SSY greater than 10000 t ($102$ t km$^{-2}$) are annotated (a). These events have exceptionally high peak suspended sediment concentrations (SSC) (b-j).

The median event specific suspended sediment yield (sSSY) is 14.3 t km$^{-2}$ (duration-normalised sSSY: 13.9 t km$^{-2}$ d$^{-1}$), and the median event suspended sediment concentration (SSC) (streamflow weighted) 1059 mg l$^{-1}$. The largest event (2014-021, Fig. 4g) exported an estimated 22019 of suspended sediment t (sSSY: 224.5 t km$^{-2}$; duration-normalised sSSY: 54.1 t km$^{-2}$ d$^{-1}$) over nearly 100 hours, with 90 % of the sediment reaching the outlet during the first 24 hours. Two events in
August 2020 (2020-027, Fig. 4i; 2020-028, Fig. 4j) occurred consecutively. When combined, these two events exported 25179 t of suspended sediment (sSSY: 256.7 t km$^{-2}$; duration-normalised sSSY: 200.2 t km$^{-2}$ d$^{-1}$). However, most events had a sSSY of less than 50 t km$^{-2}$ (Fig. 4a).

The average event duration is about 24 hours, with 90 % of events lasting under 30 hours. The remaining events have a duration of two to three days, with six events lasting longer than four days.

**4.2   Principal component analysis of event metrics**

The PCA reduced our set of 16 event metrics to seven principal components (PCs) that explain 84 % of the variance (Fig. 5h). Minor data gaps led us to consider a total of 959 events for the PCA and event clustering.

We find that PC1 (Fig. 5a) is strongly tied to streamflow and suspended sediment magnitude (i.e. mean and peak event SSC, total suspended sediment mass exported, mean and peak event streamflow, and total streamflow volume), and explains 35 % of
the total variance in the data.



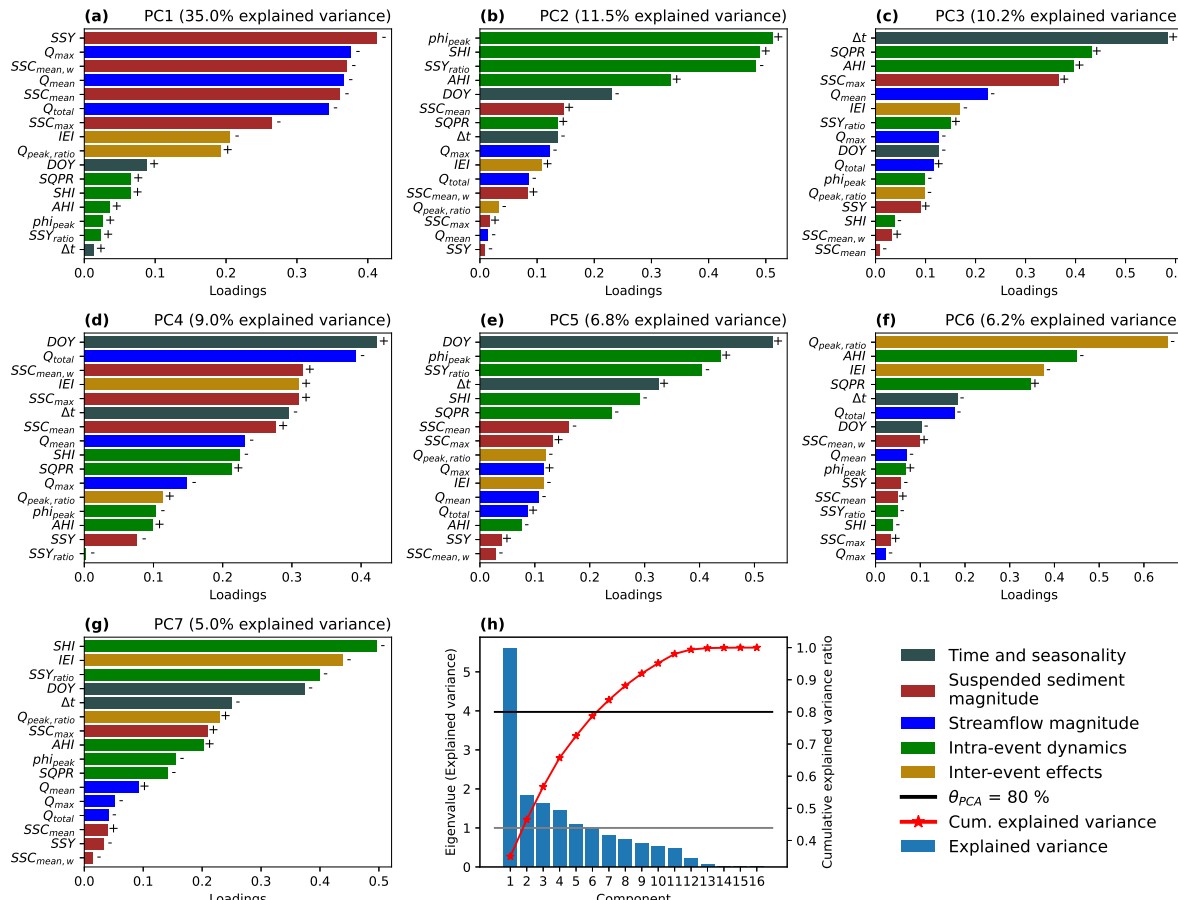

**Figure 5.** Importance of event metrics for each principal component (PC) (a-g), indicated by the sorted absolute loadings of each component. The sign of the loading, which indicates positive or negative correlation of a metric with the PC, is denoted with a "+" or "-" for positive and negative sign respectively. The first 7 PCs were chosen as these explained just over 80 % of the total variance in the data (h).

Both PC2 and PC3 relate to metrics of intra-event dynamics; PC2 (Fig. 5b) expresses hysteresis effect and direction. Higher absolute values indicates stronger hysteresis (confirmed by visual inspection of hysteresis pattern). The sign of PC2 indicates hysteresis direction, where positive (negative) values indicate clockwise (anti-clockwise) hysteresis. PC2 is essentially a combined hysteresis index of SHI, AHI, $\phi_{peak}$ and $SSY_{ratio}$. PC3 relates to the complexity of the event shape, and the similarity between hydro- and sedigraphs, expressed by mixed loadings of event duration, the ratio of SSC to streamflow peaks, SQPR, and SSC peak magnitude (Fig. 5c). Visual inspections of event hydro- and sedigraphs confirm that events with high PC3 values have multiple peaks or complex hysteresis patterns, whereas low PC3 values indicate more uniform event shapes with synchronous hydro- and sedigraphs.



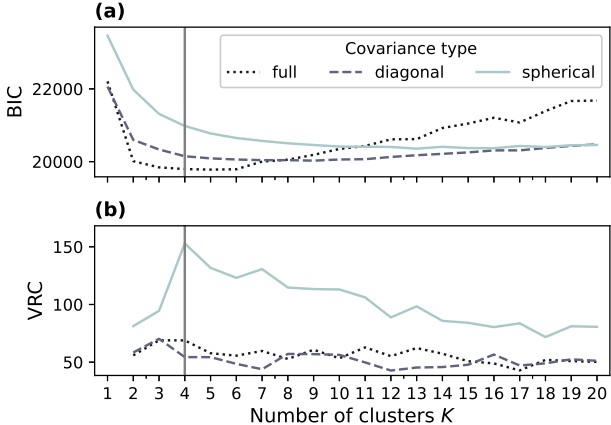

**Figure 6.** Selection of optimal number of clusters $K$ and GMM covariance type based on the Bayesian Information Criterion (BIC) and the Variance Ratio Criterion (VRC). Low (high) BIC (VRC) indicates better defined clusters. The selected number of clusters ($K = 4$) is indicated with a grey vertical line.

PCs 4 and 5 mainly represent seasonal-dependent effects. PC4 (Fig. 5d) captures the seasonal-dependent relationship be-
tween event SSC and streamflow volume, with the tendency of higher SSC for a given total streamflow volume later in the year. Events with high PC4 values tend to occur later in the season, therefore the positive correlation with DOY (Fig. 5d). PC5 (Fig. 5e) relates to seasonal- and duration-dependent hysteresis, with mixed loadings of seasonal timing (DOY), peak phase difference, $SSY_{ratio}$, and event duration.

PC6 represents mostly inter-event conditions, and two metrics on intra-event dynamics, AHI, and SQPR (Fig. 5f). Visual
checks confirm that PC6 expresses the tendency for more pronounced clockwise hysteresis (positive AHI) with higher IEI and $Q_{peak,ratio}$. Overall, AHI, IEI, and $Q_{peak,ratio}$ are negatively correlated with PC6. Finally, PC7 (Fig. 5g) relates to hysteresis, inter-event effects, and seasonality. PC7 is negatively correlated with seasonal timing (DOY), and has higher values for events occurring earlier in the year. However, PC7 explains only 5 % of the total variance in the data.

### 4.3 Event clustering

For the event clustering, we ran 60 different GMMs in total, with varying combinations of cluster numbers ($K = 1, 2, ..., 20$) and covariance types (spherical, diagonal, and full; Fig. 6). For $K < 4$, the BIC decays distinctly for all models (Fig. 6a). Models with diagonal and full covariance matrices have elbow points where the curve flattens abruptly at $K = 2$, whereas the BIC for spherical models decays smmothly with increasing $K$. For fewer clusters the more flexible covariance types (diagonal and full) have better BIC scores. Strictly judging by the BIC, the best model would be that with full covariance and $3 < K < 6$.
However, for the VRC score, the spherical-type models consistently outperform all others (Fig. 6b), with a clear peak at $K = 4$. The BIC also indicates that this number of clusters with the spherical covariance type models is reasonable, which is why we selected this variant for our interpretation.





**Table 4.** Parameters of the GMM used to assign sediment-discharge events to clusters. The mean vector $\boldsymbol{\mu}_k$ of each cluster denotes the center of each cluster's Gaussian in each dimension, i.e. for each PC. The variance describes the spread of the cluster's Gaussian. Due to the spherical covariance type, the variance is the same in all dimensions (i.e. for each PC). The mixture weights $\pi_k$ describe the relative frequency of each event cluster, which is reflected in the number of events in each cluster.

| | Mean vector $\boldsymbol{\mu}_k$ | | | | | | | Variance $\Sigma_k$ | Weight $\pi_k$ | Number of events |
|---|---|---|---|---|---|---|---|---|---|---|
| | PC1 | PC2 | PC3 | PC4 | PC5 | PC6 | PC7 | | | |
| *Cluster 0* | -3.40 | -0.11 | 1.02 | 0.77 | 0.27 | 0.12 | 0.36 | 1.56 | 0.115 | 110 |
| *Cluster 1* | -0.79 | -0.00 | -0.56 | -0.29 | -0.12 | -0.11 | -0.05 | 0.61 | 0.495 | 486 |
| *Cluster 2* | 2.13 | 0.19 | 1.20 | -0.25 | -0.22 | -0.14 | 0.21 | 3.75 | 0.193 | 176 |
| *Cluster 3* | 1.89 | -0.12 | -0.37 | 0.53 | 0.36 | 0.34 | -0.28 | 0.60 | 0.196 | 187 |

## 4.4 Detected event clusters

The model assigned 486, or about half of all detected sediment-discharge events, to cluster 1 (Tab. 4). Clusters 2 and 3 each
contain about a fifth of events (Tab. 4). Cluster 0 is smallest with 110 events, or 11 % of the total (Tab. 4). In general, clusters
1 and 3 have narrower event features ranges (Fig. 7b;d), due to the low variance of these clusters (Tab. 4). In contrast, clusters
2 and 0 have higher variances. In particular, cluster 2 has a high spread across all PCs (Fig. 7c).

The clusters have different seasonal timing (Fig. 8b). Events of clusters 0 mainly occurred from July and August, when
sediment flux was high (see average daily sSSY Fig. 4a). Similarly, cluster 1 events almost exclusively happened during
the high-flow period from mid-June to August (see average daily streamflow Fig. 4a). Cluster 3 is confined to August and
September, with a few events having occurred in July. Cluster 2 has the least seasonal signal, with some more events in May-
June and September-October.

Event magnitude (PC1) is the most prominent feature separating the clusters (Fig. 7a-d, Fig. 8a). Cluster 0 has the largest
magnitude events in terms of event SSY, peak and average SSC and streamflow, followed by clusters 1 and 3; cluster 2 has the
largest magnitude range (Tab. 4), containing the smallest and up to medium events (Fig. 7a-c).

Hysteresis effect and direction (PC2) are negligible for separating the clusters (Fig. 7a-d), with their centers being close to
the zero mean of PC2 (Tab. 4); clear differences in hysteresis are elusive (Fig. 7m-p). Judging from the SHI (Fig. 8c), cluster 0
has a slight tendency towards anti-clockwise hysteresis, and cluster 2 towards clockwise hysteresis.

Event shape complexity (PC3) separates clusters 0 and 2 from 1 and 3 (Fig. 7a-d; 8d; Tab. 4). The former have more
dissimilar event shape within the clusters, while the latter are mostly single peak events with steeply rising and slowly falling
hydro- and sedigraphs (Fig. 7e-l). Clusters 1 and 3 display little to no hysteresis compared to the other two clusters (Fig. 7m-p;
8c).



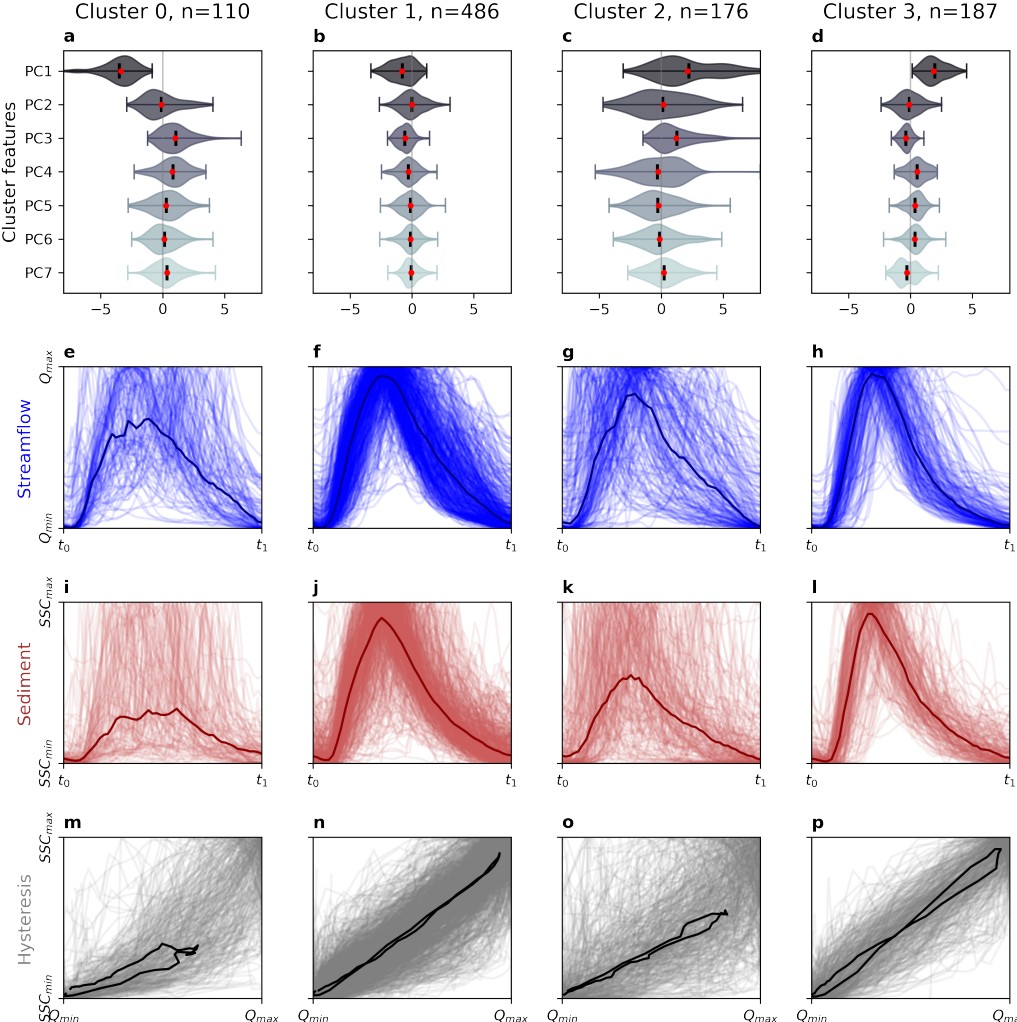

**Figure 7.** Distribution of event characteristics, i.e. PCs, for each cluster (a-d). Red dots show cluster GMM mean. Hydrographs (e-h), sedigraphs (i-j), and hysteresis pattern (m-p). Event streamflow (Q) and SSC have been normalised in magnitude and event length to enable a comparison of the event shape. All the events in the cluster have been plotted on top of each other, and the mean event cluster shape is shown in a darker line.

The seasonal dependence of the SSC-streamflow volume relationship (PC4) and hysteresis (PC5) separates cluster 0 and 3 from 1 and 2. The effect of PC4 on clusters 1 and 3 can also be seen in that (1) both events of clusters have similar average
SSC, but cluster 1 events have higher streamflow volumes (Fig. 8e), and (2) cluster 3 events occurred later in the year than cluster 1 events (Fig. 8b).



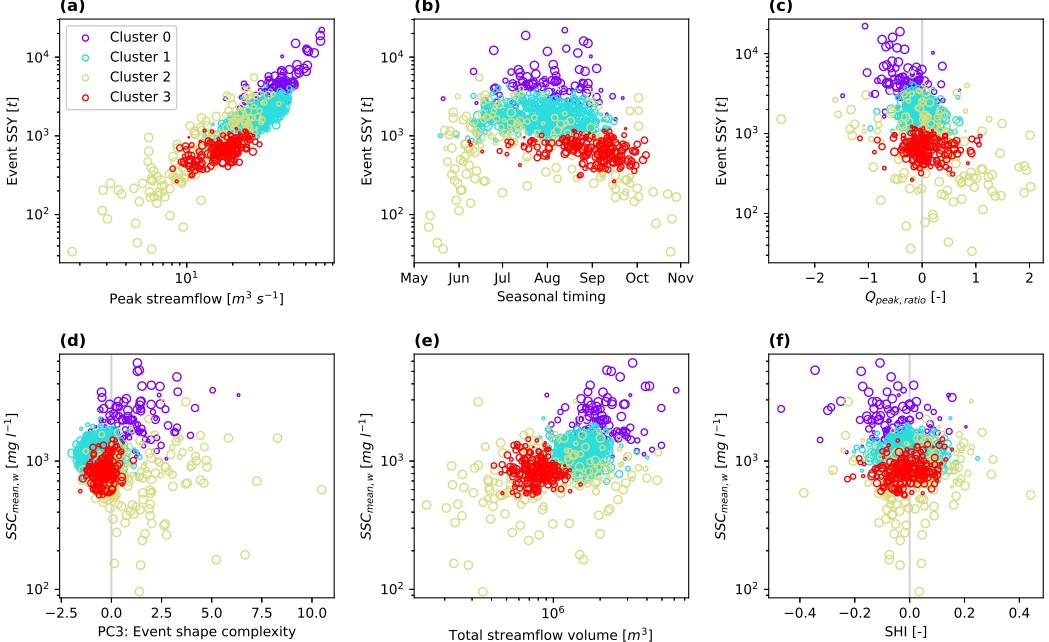

**Figure 8.** Selected sediment-discharge event metrics colored by event cluster, showing event SSY plotted against peak streamflow $Q_{max}$ (a), seasonality (b), and peak streamflow ratio $Q_{peak,ratio}$ (c), and mean streamflow-weighted SSC against PC3 (d), total streamflow volume (e), and the SHI (f). Dot size is proportional to the logit-transformed probability that an event belongs to its assigned cluster. Each cluster occupies a specific magnitude range (a-c; e). Clusters 0 and 2 have higher event shape complexity compared to clusters 1 and 3 (d). There is little difference in the hysteresis between the clusters (f).

Inter-event effects (PC6) also separate cluster 0 and 3 from 1 and 2. Clusters 0 and 3 with more positive values PC6 values, occur later in the year (Fig. 8b) and have lower $Q_{peak,ratio}$ (Fig. 8c), which means that events of these clusters tend to have higher streamflow peaks than their preceding events.

## 5   Discussion

### 5.1   Interpretation of event types

The purpose of our clustering was to find groups of events with shared characteristics but without known labels. The detected clusters form stepping stones for our objective to identify event types that relate to specific hydrometeorological forcings, catchment states, antecedent conditions, and hydro-geomorphic processes. In the following, we discuss the relevant drivers of each cluster (Fig. 9), and infer characteristic event types summarised in Tab. 5.

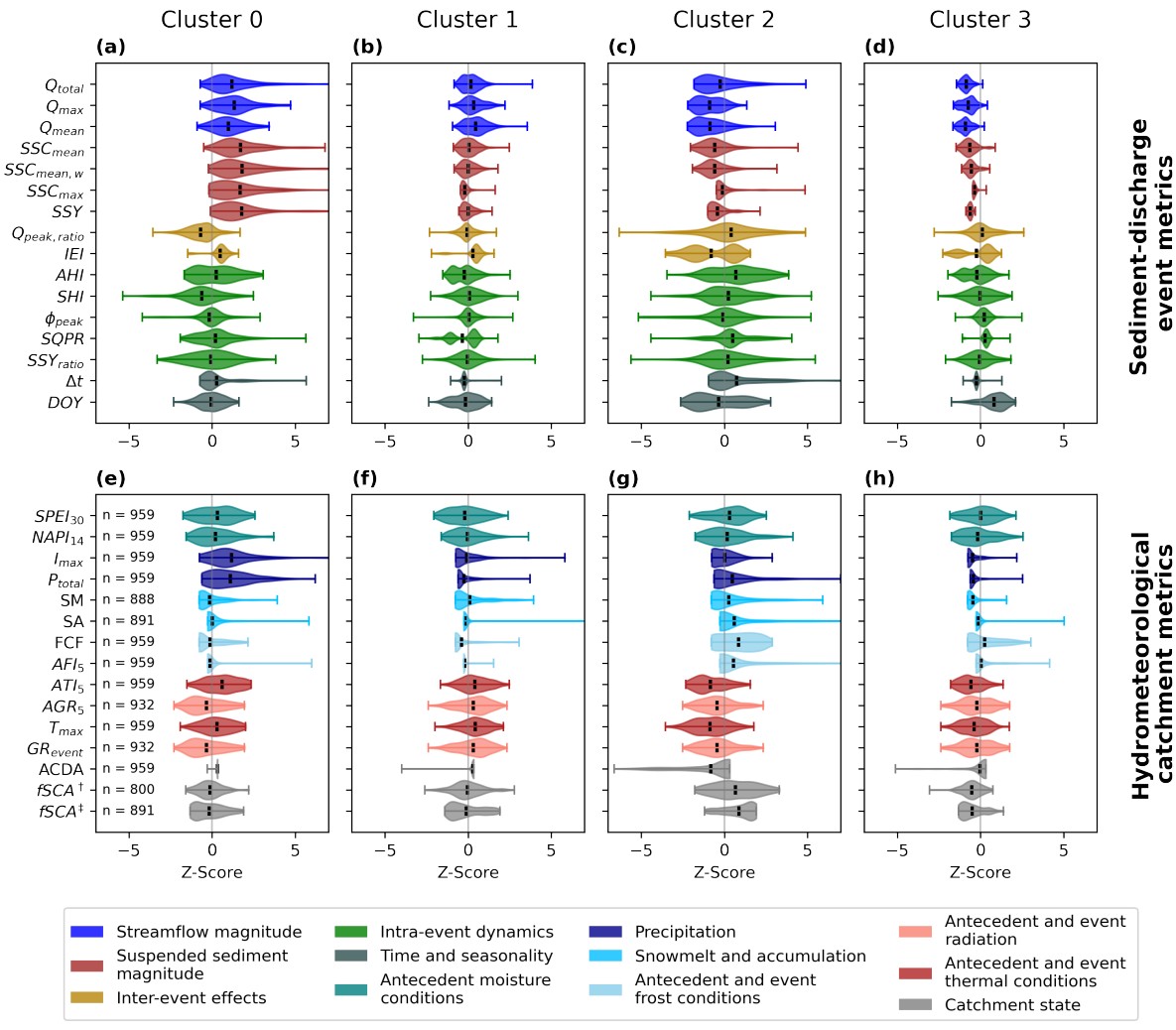

**Figure 9.** Comparison of sediment-discharge event metrics (Tab. 2) and hydrometeorological catchment metrics (Tab. 1; 3) across clusters. Coloured violins show the distributions the standardised metrics; black vertical bars are cluster means. The zero line shows the mean of each metric; units are standard deviations. Event metrics (a-d) were used to cluster events into groups. Different sample numbers for each catchment metric (e) arise from small data gaps.

### 5.1.1 Type 0: Compound rainfall-melt extremes

Event type has a median (maximum) event precipitation intensity of 7 (36) $\mathrm{mm\,h^{-1}}$ and median (maximum) total event precipitation of 15 (69) mm, and overall both are higher than for other events (Fig. 9e). As event temperatures are high, we interpret rainfall to be a key hydrological driver of this event type. The highest suspended sediment loads in mountain rivers are



**Table 5.** Summary of event types based on their sediment-discharge characteristics, antecedent conditions, driving processes, and sediment sources. Refer to sections 5.1.1-5.1.4 for detailed explanations.

| Cluster | 0 | 1 | 2 | 3 |
|---|---|---|---|---|
| **Event type** | **Compound rainfall-melt extremes** | **High melt events** | **Freezethaw-modulated events** | **Late season glacier melt events** |
| *Seasonality* | Mid-season (July-August) | Mid-season (June-August) | Early (May-June) and late (September-October) season | Late season (August-October) |
| *Events per year* | 7 (1 ... 12) | 30 (11 ... 50) | 11 (1 ... 24) | 12 (1 ... 22) |
| *Magnitude*[†] | Large | Medium | Small to medium | Small |
| | 33 (10 ... 174) $\mathrm{t\,km^{-2}\,d^{-1}}$ | 17 (7 ... 40) $\mathrm{t\,km^{-2}\,d^{-1}}$ | 7 (0.1 ... 44) $\mathrm{t\,km^{-2}\,d^{-1}}$ | 7 (3 ... 14) $\mathrm{t\,km^{-2}\,d^{-1}}$ |
| *Shape* | Complex | Synchronous, generally single peak | Complex | Synchronous, generally single peak |
| *Antecedent conditions* | Warm, low solar radiation | Warm, high solar radiation | Coldest, often below freezing | Cool |
| *Surface conditions* | Whole catchment unfrozen, some snow-cover | Whole catchment unfrozen, some snow-cover | High snowcovered area, parts of catchment frozen ($<0\,^{\circ}$C) | Low snowcovered area, parts of catchment frozen ($<0\,^{\circ}$C) |
| *Driving processes* | Erosive rainfall, snow- and glacier melt | Glacier melt and seasonal snow melt | Snowmelt and precipitation modulated by freezethaw | Glacier melt |
| *Sediment source* | Slope wash, subglacial discharge, channel erosion | Snowmelt erosion, subglacial discharge | Snowmelt erosion, slope wash | Subglacial discharge |

[†] within the range of detected sediment-discharge events

often associated with rainfall (Beylich et al., 2017; Lenzi and Marchi, 2000; Pagano et al., 2019; Rainato et al., 2021; Li et al., 2021; Wulf et al., 2012; Leggat et al., 2015; Francke et al., 2008), that trigger slope wash, mass wasting, and channel erosion (Beylich et al., 2017). The amount of sediment exported during events of this type exceeded those of other event types (Fig. 9a), involving between 17 to 224 $\mathrm{t\,km^{-2}}$ of suspended sediment (duration-normalised sSSY: 10 to 174 $\mathrm{t\,km^{-2}\,d^{-1}}$). This finding is consistent with observations from other mountain rivers, where rainfall is a key hydrological driver of the highest sediment loads, except for extreme erosion events such as natural dam failures (Korup, 2012; Cook et al., 2018), or glacier detachments (e.g. Kääb and Girod, 2023).

Conversely, only 12 out of the 110 events in this type occurred during minimal or no precipitation (<1 mm). Instead, the streamflow for these events stemmed from a combination of snow-melt (ranging from 1.2 to 12 mm) and glacier melt. We





deduce the latter based on the higher-than-average event global radiation during these events. These 12 events are similar in size to others of this type, suggesting that the magnitude of this event type is not solely determined by precipitation amount or intensity.

Meltwater influence is another likely hydrological driver of event type 0. Given that type 0 events mainly took place during the high melt season from July to August, glacier melt contributed at its most to streamflow. For a sub-catchment of Vent-Rofental the glacier melt contribution to streamflow is estimated at 40 to 70 % (Schmieder et al., 2018). For earlier events in May and June, snowpack melting also adds to streamflow, between 35 and 80 % (Schmieder et al., 2018). Our assumption of significant meltwater contribution to type 0 events is corroborated by high maximum temperatures and an above-average antecedent thawing index (Fig. 9e), both indicating snow- and glacier melt (Hock, 2005; Woo, 2005). Thus, we attribute the high streamflow peaks and volumes of this event type to the combined effect of meltwater and rainfall-runoff.

The combination of rainfall, and snow or glacier melt as driving processes may also explain the high SSY of type 0 events. During high glacier melt, subglacial sediment discharge likely boosted suspended sediment load (Costa et al., 2018; Delaney et al., 2018b; Zhang et al., 2022). Similarly, snow-melt runoff and erosion can contribute sediment-rich meltwater to the streams (Wu et al., 2018; Lundekvam and Skøien, 1998; Costa et al., 2018). High antecedent thawing index (Fig. 9e) indicates subglacial sediment discharge in the days prior to the events, and thus an already high ambient SSC. Large portions of the catchment are prone to rainfall erosion during type 0 events, as indicated by a large actively contributing drainage area (ACDA) and low snowcover (Fig. 9e). The high streamflow rates in type 0 events likely induce channel bank erosion, particularly in proglacial areas with unconsolidated sediment, thus further elevating SSC.

Multiple hydrological drivers and multiple sediment sources explain the complex event shapes of this cluster (Fig. 8d; 7e). Similar complex event shapes were reported after rainstorms in proglacial areas (Leggat et al., 2015; Orwin and Smart, 2004). A visual inspection of event hydro-, sedi- and hyetographs shows that several events had multiple peaks attributable to different melt or rainwater pulses, i.e. a glacier melt-driven sediment pulse, with a superimposed rainfall-driven pulse, e.g. during events 2006-016 (Fig. 3b), 2010-024 (Fig. 3c), 2012-48 (Fig. 3e), 2018-020 (Fig. 3h), and 2018-027 (Fig. 10). Mass wasting that abruptly delivers large amounts of sediment to the channels may explain the complex event shapes and exceptionally high SSC peaks of some type 0 events. This happened during the consecutive events 2020-027 and 2020-028 (Fig. 4i-j), when a debris flow impacted the tip of the glacier Hintereisferner. The subsequent fluvial reworking of the deposits elevated SSC for several days.

### 5.1.2 Type 1: High melt events

The main hydrometeorological drivers of this event type are glacier and seasonal snowpack melt. Event global radiation and maximum temperature are above average for most events of this type (Fig. 9f). Both metrics indicate glacier and snow melt (Hock, 2005; Woo, 2005), and also melt extremes (Cremona et al., 2023; Thibert et al., 2018). Most events (90%) of this type lasted between 20 and 27 hours and had single peak hydrographs (Fig. 7f; 11), consistent with the characteristic diurnal cycle of glacier melt (Hock, 2005) and seasonal snowpack melt (Schmieder et al., 2018). The events occurred mainly during the high flow period from mid-June to August, dominated by snow and ice melt in Vent-Rofental (Strasser et al., 2018). Thus, from June





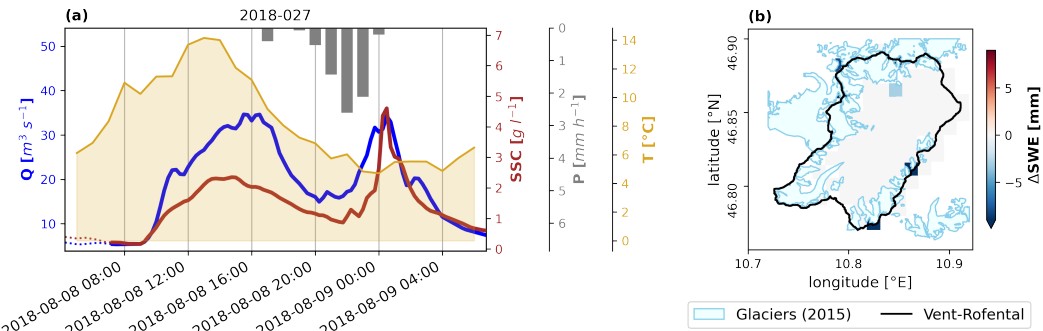

**Figure 10.** Example type 0 event. Left-hand panels show event hydrographs of 15-min streamflow (Q), sedigraphs of 15 min suspended sediment concentrations (SSC), hyetographs of hourly precipitation (P), and hourly catchment average air temperature (T). Right-hand panels show changes in SNOWGRID SWE from first to last day of event. Event 2018-027 was not influenced by snow-melt (b). An initial pulse of glacier melt during the day, was followed by a rainfall-driven pulse at night (a).

to early July, type 1 events were likely driven by snowpack melting, whereas from July to early September they were likely fed by glacier melt mainly. Streamflow volumes would have responded to warm antecedent conditions and large ACDA (Fig. 9f),
as with most of the glacier and snow surfaces likely near the melting point at the event onset, more easily inducing melt (Hock, 2005; Woo, 2005).

Subglacial sediment is often the prime source of suspended loads in glacierised basins (Delaney et al., 2018a; Zhang et al., 2022). The positive correlation between annual glacier mass loss and sSSY supports this notion for Vent-Rofental (Schmidt et al., 2022). Event SSY is the result of high mean SSC and high streamflow volumes (Fig. 9b), which are elevated during
glacier melt (Costa et al., 2018). Snowmelt often results in surface runoff and erosion (Lundekvam and Skøien, 1998), as freshly thawed soils have low infiltration rates and increased surface runoff (Wu et al., 2018). In spring, sediment mobilised by snow-melt-driven runoff from hillslopes can raise SSC in mountain rivers (Costa et al., 2018; Wu et al., 2018). Thus, we infer that the suspended sediment of this event type is mainly sourced from snow-melt erosion in June to early July, and subglacially in July to early September.

### 5.1.3 Type 2: Freezethaw-modulated events

The primary hydrometeorological forcing for this event type we attribute to alternating or co-occurring freeze and thaw, as shown by significantly higher FCF (Fig. 9g). Precipitation and snow-melt have secondary roles, indicated by generally high snow-melt rates and the second-highest precipitation rates (Fig. 9g). Type 2 events happened during the coldest periods, and with the most extensive snow cover. They occurred throughout the season, but mostly in May-June and September-October.
Similar to event type 0, the complex event shape and high variability in sediment-discharge characteristics within this event type (Fig. 9c) reflect multiple drivers of streamflow and SSC. Unlike type 0 events, the complex event shape of type 2 events



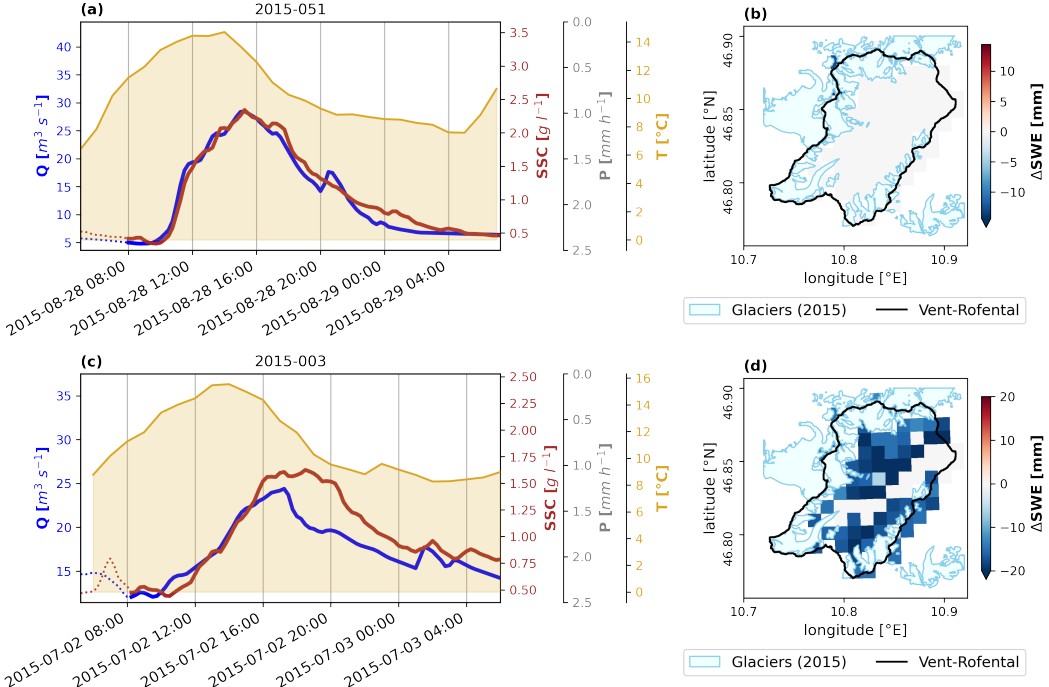

**Figure 11.** Examples of type 1 events. See explanation Fig. 10. Event 2015-051 represents a glacier-melt-induced event (a), with no apparent influence of snow-melt (b). Event 2015-003 (c) is mainly snow-melt-driven (d). Note the high catchment temperatures for both events.

partly result from a modulating effect of freezethaw on the driving hydrological processes. A similar effect of catchment freezethaw state regulating suspended sediment dynamics has been proposed for the Tibetan Plateau (Li et al., 2021).

From examining event hydro-, sedi- and hyetographs, we infer that three different mechanisms may have generated type 2
events: (a) snow-melt (Fig 12a-b), (b) rain on snow (Fig 12c-d), and (c) precipitation (Fig 12e-f). Many events arose from a combination of some or all of these mechanisms, with temperature and freezethaw modulating both magnitude and dynamics. This modulating effect mainly lowers snow-melt rates, either due to freezing that prevents snow-melt, or due to cold precon- ditions that steer the energy balance to warming initially rather than melting the snowpack (Woo, 2005). For precipitation, the colder conditions regulate the frozen to liquid precipitation ratio either *spatially*, i.e. higher elevations accumulate snow while
lower elevations generate runoff, or *temporally*: precipitation falls as rain initially, turning to snow later as the catchment cools to freezing during passage of a cold front.

### 5.1.4   Type 3: Late season glacier melt events

Events of this type have several sediment-discharge characteristics resembling those of the glacier and snow-melt event type (type 1), including a single peak, synchronous hydro- and sedigraphs, and durations of about one day. What distinguishes type



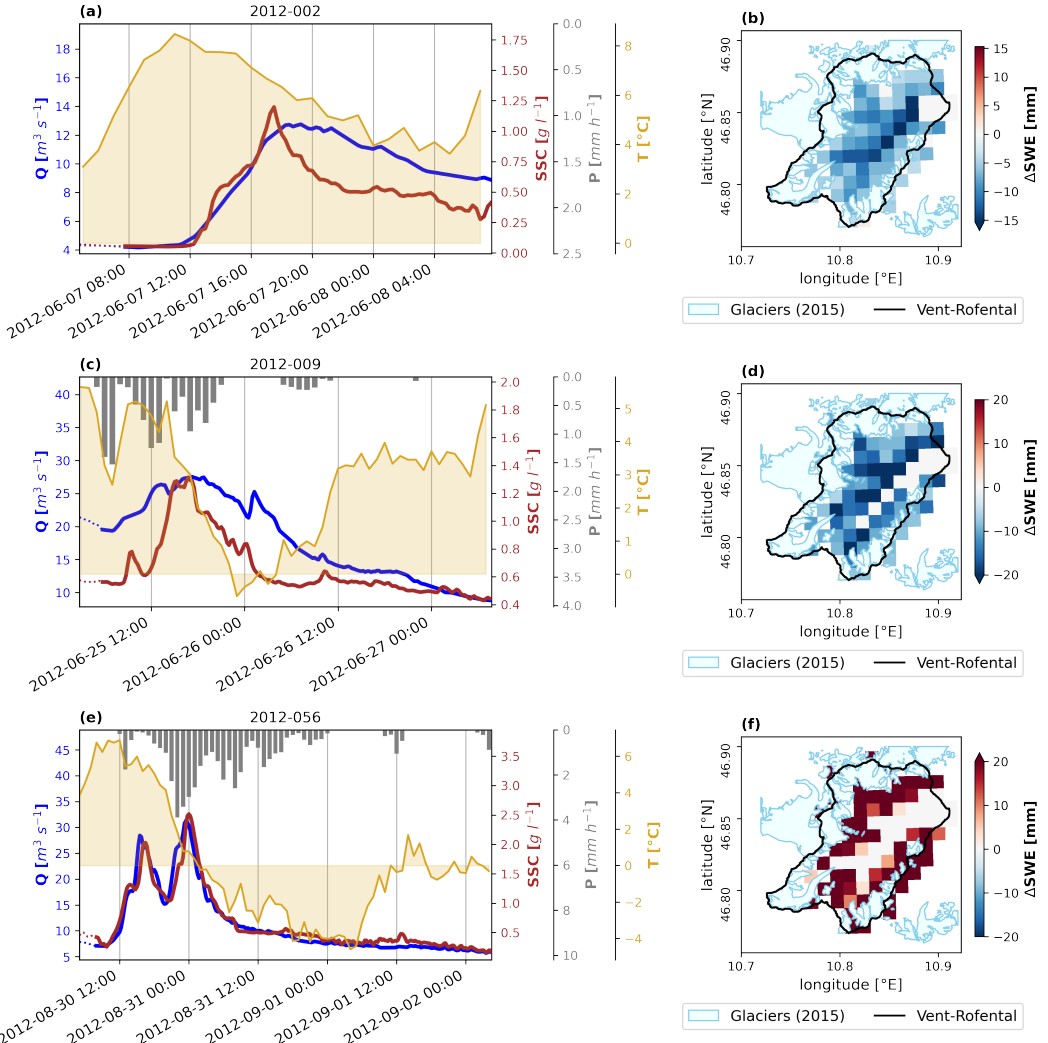

**Figure 12.** Examples of type 2 events. See explanation Fig. 10. Event 2012-002 (a) is an example of an early season, smaller magnitude snowmelt event where the lower elevations contributed most (b). Event 2012-009 is a rain-on-snow event where both snow-melt (d) and rainfall (c) contributed to streamflow. Event 2012-056 is a precipitation-driven event (e), with two initial rainfall-driven pulses, and subsequent snowfall as the catchment froze (f).

3 events are lower streamflow and SSY magnitudes, and their timing in the late melt season, with 80% having occurred after mid-August.

The main drivers of type 3 events appear to be unrelated to snow-melt or rainfall, given that they are happened during little to no precipitation or snow-melt, and during the lowest overall snow cover. Yet, the similarities in event characteristics, coupled with apparent thermal influences, lead us to propose that type 3 events are linked to late-season glacier melt. Following our





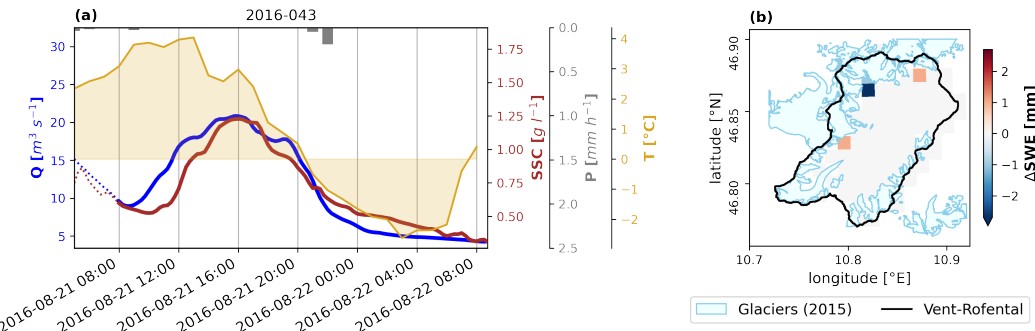

**Figure 13.** Example of type 3 event. See explanation Fig. 10. Event 2016-043 is a small magnitude glacier melt event due to low and below freezing temperatures (a) and no snow-melt (b).

argumentation for type 1 events, we posit that subglacial sediment discharge is the primary source of suspended sediment for type 3 events.

The low streamflow magnitudes in type 3 events can be attributed to colder antecedent conditions and lower temperatures. Low ACDA values further indicate that parts of the catchment were below freezing, likely higher up near the glaciers. Hence, initially the energy from global radiation was primarily directed towards heating the ice to the point of initiating melt (Hock, 435   2005), resulting in lower melt rates. Alternatively, it is possible that only the lower parts of the glaciers were warmed sufficiently to contribute meltwater to the streams.

While the mean SSC of type 1 and type 3 events are comparable (Fig. 8d-f; 9b,d), the significantly lower streamflow magnitudes limit the overall SSY of type 3 events (Fig. 8a-c; 9b,d). Hence, the low SSYs of type 3 events might reflect transport-limited conditions rather than decreasing availability of subglacial sediment throughout the melt season, as observed 440   elsewhere (Delaney et al., 2018b).

## 5.2   Event type contributions to annual suspended sediment yield

We find that a large portion of annual sSSY in Vent-Rofental is exported in a few high-flow events, consistent with previous studies of fluvial sediment dynamics (Gonzalez-Hidalgo et al., 2013; Vercruysse et al., 2017). On average, 40 % of the annual SSY is exported in 2 % of the time in Vent-Rofental (Schmidt et al., 2022). Recalling that our event definition only included 445   SSC above the 90th-percentile, a substantial part, on average 36 %, of a given year was occupied by events (Fig. 14d).

On average, the four identified event types involved 83 % of annual sSSY (Fig. 14b), although they covered only 37 % of the time. Despite their large magnitude, as rainfall-melt compound extremes (type 0) were rarest (Tab. 5: Fig. 14d), they had the only second largest share (25.6 %) of annual sSSY (Fig. 14b). The most frequent high melt events (type 1) contributed the largest portion (41.6 %) of annual sSSY (Tab. 5: Fig. 14d). Types 0 and 1 thus dominated the annual sSSY, showing the 450   relevance of glacier melt and erosive rainfall for suspended sediment fluxes in our study area. A similar influence of these



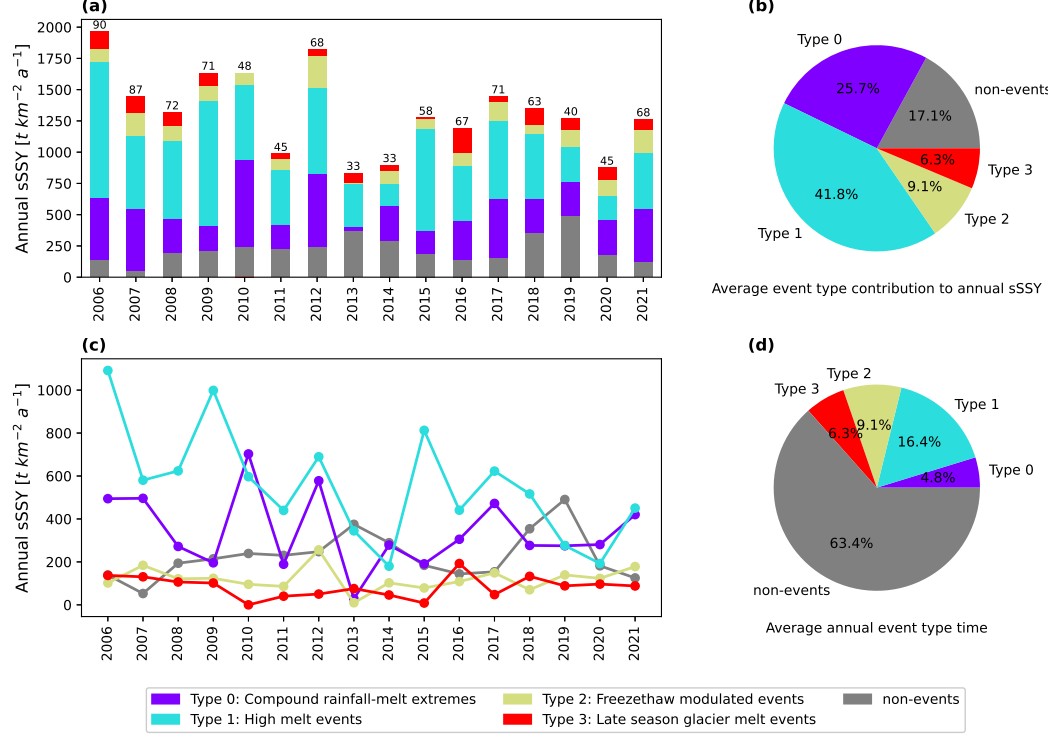

**Figure 14.** Annual specific suspended sediment yield (sSSY) contributed by each event type (a), with considerable inter-annual variability. The total number of events each year is annotated above the bars. Average contribution of each event type (b) shows the dominance of type 0 and type 1 events. The sSSY attributed to type 1 events declines during the study period (c), with the contributions for the other types remaining more constant. The average proportion of time during the observation season from May-October for each event type (d) is rather small compared to the time where no events occurred.

hydrometeorological drivers on suspended sediment transport has been reported for the heavily regulated upper Rhône Basin in the Swiss Alps (Costa et al., 2018), and the maritime mountains of Western Norway (Beylich et al., 2017).

High melt event annual sSSY contribution declined during our study period (Fig. 14c). A recent reconstruction of suspended sediment export from Vent-Rofental suggests a negative trend in mean annual sSSY of -7.6 t km$^{-2}$ a$^{-1}$ after a change point in 1981 (Schmidt et al., 2023b). Thus we may be seeing declining influence of subglacial sediment discharge on SSC as the glaciers in Vent-Rofental shrink and disappear.

Annual SSY is projected to continue to decrease until 2100 in Vent-Rofental, even when inflating yields from days with heavy precipitation (Schmidt et al., 2023a). This fits with our findings that high yield events have erosive rainfall as an important driver on top of a high base SSC from subglacial sediment discharge. Thus, future potential increases in SSC driven by erosive rainfall may be offset by decreasing glacier melt contributions to SSC.

455

460





### 5.3 Benefits and limitations of methodology

Our approach enables event-based analysis of river sediment fluxes, by detecting, characterising and grouping similar sediment-discharge events. These groups can then be interpreted to understand under which conditions episodic sediment fluxes occurred.

The main advantage of this clustering method is that it does not require any prior knowledge or labels (Tarasova et al., 2019). Our methodology is transferable and can be used in any catchment with sub-daily time series of streamflow and SSC. Our approach is also customizable, as the parameters for event detection (search window and SSC threshold) are easily adapted to fit other catchments. The use of PCA condenses the input data to the most informative event characteristics in the clustering process, while information criteria help to identify the optimal cluster model and number of event types. Another strength of our approach is the GMM itself, as we can examine the probabilities of belonging to the assigned cluster. Thus we can check how reliably the GMM assigns events to each of the clusters, which cluster algorithms such as K-medoids or hierarchical clustering commonly used (e.g. Leggat et al., 2015; Javed et al., 2021; Mather and Johnson, 2015) are unable to do.

The drawback of using a data-driven approach such as clustering is that the derived event typology is sensitive to the choice of sediment-discharge metrics. Both the PCA and GMM can only learn from the data it is trained on. Moreover, data-driven approaches can be sensitive to sample size. Although we have not explicitly tested how sensitive our results are to the metric selection and amount of data, the inclusion of an additional year of data did not substantially alter the number or characteristics of event clusters or types.

The four event clusters detected in this study represent groups of events, each with distinct sediment-discharge characteristics. Two points give confidence that the detected event clusters are not random groupings. First, the relative frequencies of events are not equal, i.e. the events were not split into 4 equal portions. Second, we included the possibility of having no clusters when testing the different GMMs (i.e. $K = 1$).

How consistent event typologies derived with our approach are across various catchment types and regions requires further research and testing. Hence, we refrain from proposing a universal suspended sediment flux typology from this study of a single catchment. Rather we hope that our approach can serve as a knowledge discovery tool to identify catchment-specific event types in all types of hydrological basins.

In general, our approach has high data requirements as both high temporal resolution and long temporal extents are necessary to gather enough events for clustering, especially when dealing with the high variability typical of sediment transport. In order to interpret the detected event clusters a substantial amount of data representing hydrometeorological forcings, catchment state and antecedent conditions is needed.

### 5.4 Implications of findings and outlook

Event magnitude is rarely considered in sediment-discharge event typologies. Most other studies focus primarily on discharge-SSC hysteresis, and other intra- and inter-event dynamics (e.g. Tsyplenkov et al., 2020; Haddadchi and Hicks, 2021, 2020; Hamshaw et al., 2018). Classifications based on hysteresis degree and direction are most commonly used. Yet, our study shows that magnitude is the most important characteristic, accounting for a third of the event regime variability and being the main





factor distinguishing the event types. Our findings emphasise that hysteresis only explains about a tenth of the variability in the
events, and it is not an important feature for distinguishing the event types. This highlights the methodological challenges of
hysteresis pattern analysis, namely its difficult and highly context-driven interpretation due to feedback mechanisms and inter-
actions between multiple drivers (Vercruysse et al., 2017). While hysteresis analysis can be a useful tool (Malutta et al., 2020),
it may be insufficient for event type classification in alpine catchments if multiple drivers of suspended sediment transport are
concerned.

We find event shape complexity, i.e. degree of hydro- and sedigraph synchronicity, to be a key feature for distinguishing
event types. In our study area higher complexity generally indicates multiple processes generating streamflow and suspended
sediment. We were able to capture this effect, despite only implicitly including the event shape in the clustering though metrics
like SQPR. The hydro- and sedigraph synchronicity is somewhat included in hysteresis indices, as values close to zero indicates
this. METS clustering is an approach that uses event shape explicitly to cluster events (Javed et al., 2021). However, they do not
include magnitude, which we find to be the most important characteristic both for describing the event regime and for clustering.
An important implication of our results are that multiple processes can generate the same signal in sedi- and hydrographs. Care
should be taken when interpreting from these alone, e.g. hysteresis classifications connected with processes.

  We link the sediment-discharge characteristics of each event type with hydrological and geomorphic processes, although
how clear this linkage is varies with each event type. For instance, the Gaussians of type 1 and 3 cover a more defined variable
space, and their distinct diurnal hydro- and sedigraphs was attributed to snow- and glacier melt. For type 0 and 2 events,
the attribution to specific processes is less clear, as multiple hydrological drivers are involved and the variability within the
clusters are higher. Therefore, care should be taken when interpreting the results clustering-based approaches, as the events are
grouped based on the observed effect of hydrological and geomorphic processes rather than the processes themselves, as with
a process-based approach.

## 515 6 Conclusions

Identifying drivers of suspended sediment dynamics in mountain rivers remains challenging. Nonetheless, event-based analysis
of fluvial suspended sediment transport can shed light on the main catchment processes and conditions that drive suspended
sediment fluxes. We present an approach for identifying event types based on automatic event detection and clustering in the
high alpine Vent-Rofental catchment in the Ötztal Alps, Austria.

Magnitude and event shape complexity (hydro- and sedigraph synchronicity) are the main characteristic that define the event
types in our study area. Rainfall is important for triggering extreme events, but their magnitude is due to a compound effect
from subglacial sediment discharge, slope wash, and fluvial channel erosion. Sediment-discharge hysteresis is important for
characterising the overall event regime, but not for distinguishing event types. Rather event shape complexity (hydro- and
sedigraph synchronicity) is the second most important factor after magnitude for separating event types.

We connect the four identified event types to different drivers. Extreme compound rainfall-melt events (type 0) are driven
by a compound effect of high subglacial sediment export and ice melt, erosive rainfall, channel erosion, and possibly rainfall-

triggered mass wasting. High melt events (type 1) are driven by snow-melt and erosion from June to early July and subglacial sediment discharge and glacier melt from July to early September. Freezethaw-modulated events (type 2) are driven by snow-melt, rain-on-snow, and colder precipitation events involving snowfall. Late season glacier melt events (type 3) are driven by

subglacial sediment discharge and glacier melt, with cooler event and antecedent conditions.

Differences in antecedent moisture are vague for the different event types. The two higher magnitude event types, type 0 and 1, have distinctly warmer antecedent conditions compared to the smaller magnitude event types, type 2 and 3.

In total, events contribute the bulk of annual suspended sediment yields (SSY) (on average 83 %), whereby the extreme compound rainfall-melt events and high melt events contribute the most, more than half of annual SSY. The Vent-Rofental

represents a high alpine catchment, were glacier melt and subglacial discharge in conjunction with erosive rainfall constitute the primary drivers of event-scale suspended sediment dynamics.

We offer a first approach for deriving sediment-discharge event typologies, designed to be transferable and adaptive. Our findings highlight the importance of multiple geomorphic and hydrological processes driving event-scale suspended sediment dynamics in mountain rivers, and that events are relevant for annual yields. Future studies could use our findings to inform

process-based modelling and classification approaches.

*Code and data availability.*    Streamflow and suspended sediment concentration time series are available at daily resolution for download from the data portal of the Hydrographic Service Austria https://ehyd.gv.at and at 15-min resolution upon request from the Hydrographic Service of Tyrol via wasserwirtschaft@tirol.gv.at. All datasets by GeoSphere Austria are freely available for download from their online data portal https://data.hub.geosphere.at. The MODIS snowcover maps version v1.0.2 by Matiu et al. (2019) can be downloaded here: https://doi.org/10.

5281/zenodo.3601891. The code and scripts used to detect, characterise and cluster events are available as a python module https://github.com/skalevag/hysevt/releases/tag/v0.1. The main results of this study, namely the detected events, calculated event and catchment metrics, event characteristics from the principal component analysis, identified event clusters, event type annual suspended sediment yields, and event plots of all events in each cluster, can be found in this data repository: https://doi.org/10.23728/b2share.4806f852f25541b4a3206b0d8110907c

*Author contributions.*    AS developed the general idea and conceptualised the study, with input from AB and OK. AS developed the method-

ology and performed the formal analysis, under supervision of AB and OK. AS prepared the original manuscript draft, including all the figures, and AB and OK critically reviewed, commented and revised the manuscript.

*Competing interests.*    The authors declare that they have no conflict of interest.

*Acknowledgements.*    Lena Katharina Schmidt helpfully provided her extensive insights and experience of the sediment dynamics in the study area, Vent-Rofental, and was instrumental in developing the inter event index (IEI).



ChatGPT 3.5 was used in the editing process of the results and discussion sections. It was only used to suggest rephrasing of existing text, mainly single sentences or paragraphs. This applies to roughly 5 % of the aforementioned sections. Not at any point in the manuscript writing and editing process was ChatGPT used to generate new text or ideas, or to summarise existing concepts and literature.





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
