# Peer review of "Inferring sediment-discharge event types in an Alpine catchment from sub-daily time series"

_Hydrology and Earth System Sciences, 2023_

## Author Response (AR1)

**Authors' reply to Anonymous Referee #1 from 26 Jan 2024**
https://doi.org/10.5194/hess-2023-300-RC1

**General comments:**

This manuscript examines sediment discharge events in an Alpine catchment to examine the type of events that typically occur given hydrological and sediment transport conditions. I appreciate the author's interest in this subject and their thorough approach to this work.

In my opinion, the ideas and findings presented in this manuscript are highly relevant as hydrology, and thus sediment transport conditions, evolve in Alpine environments. While I found the methods used to examine these events generally robust, there are also some substantial questions and concerns I have about them, that I hope are resolvable.

I believe the manuscript is generally well written. However, there are some writing matters of phrasing that I believe should be clarified. Additionally, I found the way that the event types and their interpretation are presented to be slightly cumbersome and might be able to be improved. In the comments below, this is described in detail.

In my opinion, these matters should be addressed before publication. I believe that with these revisions the manuscript can likely be a valuable asset to the field and one that I may cite in my research.

My general and specific scientific comments are below. Hopefully, the editor and authors find them useful.

**General scientific comments:**

1. I am curious about the frequency role of these events changing over the study period as the catchment has experienced a substantial change in hydrology and glacier cover. Furthermore, research such as Lane and Nienow (WRR 2019) shows that characteristics, such as water discharge variability have changed in these catchments.

**Answer:** Thank you. Certainly, the catchment has experienced substantial changes in hydrology and glacier cover in recent decades (see Schmidt et al. 2022; 2023). However, given the high inter-annual variability of annual suspended sediment yields (SSY) in the catchment, the 16-year study period in this study is too short to definitively say whether the frequency of events of a specific size range has been changing, especially on an annual basis. Decadal streamflow trends in Tyrol (Kormann et al., 2014) show a seasonal shift in the timing of high flows since 1980. Yet, changes in the timing of streamflow does not affect our event detection.

We attach some figures showing the frequency of events and event types over the study period. The total number of events per year covaries with the annual SSY (attached Figure 1), whereas event types 0, 2, and 3 show no trend in their frequency over the study period (attached Figure 2). The number of type 1 events declines slightly over the study period (attached Figure 2). However, the high inter-annual variability and short study period makes this difficult to assess.

- Schmidt, L. K., Francke, T., Rottler, E., Blume, T., Schöber, J., & Bronstert, A. (2022). Suspended sediment and discharge dynamics in a glaciated alpine environment: identifying crucial areas and time periods on several spatial and temporal scales in the Ötztal, Austria. Earth Surface Dynamics, 10(3), 653–669. https://doi.org/10.5194/esurf-10-653-2022

- Schmidt, L. K., Francke, T., Grosse, P. M., Mayer, C., & Bronstert, A. (2023). Reconstructing five decades of sediment export from two glacierized high-alpine catchments in Tyrol, Austria, using nonparametric regression. Hydrology and Earth System Sciences, 27(9), 1841–1863. https://doi.org/10.5194/hess-27-1841-2023
- Kormann, C., Francke, T., & Bronstert, A. (2014). Detection of regional climate change effects on alpine hydrology by daily resolution trend analysis in Tyrol, Austria. Journal of Water and Climate Change, 6(1), 124–143. https://doi.org/10.2166/wcc.2014.099

2. In my opinion, the definition of an event and its characteristics should be more narrowly defined and likely altered. From my interpretation of the manuscript, it seems that a combination of hydrological and sediment transport conditions defines events. However, hydrological conditions, along with sediment access, drive sediment transport. Thus, should events be defined by their sediment discharge, not hydrological, characteristics? In the way I see the experiment now, it seems like one may suggest that hydrological conditions result from the sediment discharge. This should be resolved. Some of the implications of this are discussed below.

**Answer:** While we appreciate the referee's opinion, we have decided to maintain our definition of events for mainly two reasons. First, our objective was to detect events that involved not only peaked water discharge, but also high loads of sediment (Fig. 2a). Various methods exist for detecting either hydrological or sedimentary events (see our brief literature review), though these separate approaches may overlook important effects of hysteresis. Second, considering water and sediment characteristics together gives us a richer, and geomorphically more meaningful, representation of the diversity of events, making them more amenable to clustering.

3. There are some interchangeable and imprecise terms in the text. Some are noted specifically below.

**Answer:** Thank you for pointing this out. See our comments below for our suggested changes.

**Specific scientific comments:**

Title: Alpine or alpine? No opinion on my part, but recommend considering it.

**Answer:** Indeed the correct spelling according to the Oxford English Dictionary (OED) is "Alpine". We will change this in the title.

**Comments on abstract:**

Ln 9. Can some terms be simplified for a lay reader? These include "Freeze-thaw state" and "antecedent conditions."

Ln 10. What is "event shape"? I can intuit what it is, but is there a more general way to keep this in the abstract?

**Answer:** We agree that some of these terms can be simplified or clarified in the abstract. Our suggestion is to change "freeze-thaw state" to "fraction of catchment area with freezing temperatures", and "event shape complexity" to "complexity of the hydro- and sedigraphs". However, we prefer to keep the term "antecedent conditions" as is, since this is a common scientific term and alternative phrasings such as "conditions leading up to events" or "preceding conditions" is either too informal or imprecise.

Ln 14. "Higher" than what? Does it mean that snow and ice melt events drove 40% of annual yield?

**Answer:** The phrase "higher magnitude glacier and snow melt events" was intended to refer to type 2 events, and is meant contrast these events from late season glacier melt events (type 4). However, we see that the current phrasing is cumbersome and easy to misinterpret. We suggest to use the phrase "Glacier and snow melt events driven by warm conditions and high insolation".

Ln 17. Does it represent "suspended sediment dynamics"? or hydrological conditions resulting in typical sediment transport events? I somehow see events as different from processes.

**Answer:** Thank you for pointing out our unclear use of the term "sediment dynamics" here. It would be more precise to say "suspended sediment transport conditions" here, and we will accordingly make this change.

**Comments on introduction:**

Ln 63. Consider defining "shape."

**Answer:** In this context, and being consistent with the two studies cited here, the term "temporal pattern". We will change the sentence accordingly.

Ln 82. "How well" seems a bit vague. what about "Do sediment transport events of a type share hydrometeorological drivers?" Also, this seems difficult to answer in the context of the presented research as hydrology has changed over the study period (Point GSM 1 above) and I am not sure that events are separate from a hydrological driver (see GSM 2 above).

**Answer:** Thank you for this suggestion. We propose to change the research question to: "Do events of a type share certain hydrometeorological drivers?"

Still, we disagree with the referee's view that it will be difficult to analyse common drivers because of a changing hydrology during the study period. As we calculate the characteristics and indices of hydrometeorological drivers separately for each event they should not be influenced by decadal hydrological trends. The data on catchment conditions and hydrometeorological forcings that we use to assess the key drivers of each event type do carry some uncertainties, as they only give an indication of the dominant hydrological processes rather than directly measuring these directly. However, we address this issue sufficiently in the interpretation and discussion of the event types.

We agree with the referee in that events are separate from a hydrological driver. Indeed, this is the basis for our interpretation of the events and their attribution to different drivers. The central idea of our paper is that events (pulses of streamflow and SSC) must represent to some degree an integrated catchment response, and that by grouping events with similar characteristics we gain groups that share similar hydrological drivers. Hence, the similarity between events and drivers is at the level of clusters or groups.

**Comments on study area and methods sections:**

Section 2 - Please describe the amount of glacier cover. This will be important in evaluating the subglacial sources of sediment.

**Answer:** The amount of glacier cover, as of the latest glacier inventory in 2015, is 28%. This is already specified in the text, Ln 93-94.

Paragraphs 138-147 - This is at the crux of my comment on GSM 2 above. Also, I found it a bit strange to suggest that there is a sediment discharge event (Ln 138) and then speak to the hydrological component. Also, the last sentence (143-144) would omit a supply-limited event. For instance, if the stream flow was "normal" and supply-limited. Then a landslide, not associated with rainfall, deposits a large amount of sediment in the stream, would this be defined as an event? The concentration and flux would increase, potentially by a lot, but the water discharge could remain constant.

**Answer:** Our choice to use of streamflow time series to delineate events was made to be consistent with previous studies (highlighted in the indicated paragraph). However, as the referee rightly points out, the subsequent filtering of events by suspended sediment concentration (SSC) could lead to exclude supply-limited events.

Our event definition would lump a hypothetical landslide case (not triggered by rainfall) together with an unconnected streamflow pulse. Still, this process would be detected as an event, if the SSC measured at the outlet was above the $90^{th}$ percentile threshold.

It is difficult to reconcile the two hypothetical cases that the referee prosposes, i.e. (1) supply-limited conditions with peak below the SSC threshold, and (2) a suspended sediment pulse unrelated to a hydrological driver, e.g. a landslide unrelated to rainfall, into an event definition that allows for both. Removing the filtering according to event peaks would no longer be an event detection, but rather splitting the entire time series into event-sized portions.

In summary, as fluvial sediment transport is driven by hydrology, we find it reasonable to delineate events according to streamflow. Cases where SSC spikes occur without any commensurate increase in streamflow would be very rare (at least in the Vent-Rofental catchment).

Ln 150. "21 hour search window." I am uncertain about this and how this impacts the statement at Ln 156. Does this imply that there cannot be more than one minimum in 21 hours? or more than 1 event cannot occur in a 21-hour period? or that events cannot be more than 24 hours in length? This is contradicted in Figure 3. Please clarify.

**Answer:** The local minima method uses a centred 21-hour search window, meaning that if the time step at the centre of the search window is the minimum of the search window, it is identified as a local minima. Thus the closest two local minima can be is half the search window length, i.e. 10.5 hours. In our catalogue of detected events the shortest event duration is 13.25 hours. There is no theoretical maximum for the duration of an event, except the length of the streamflow time series. The longest detected event in our catalogue is 191.25 hours, i.e. just short of 8 days.

Figure 2- PCA and GMM. Please introduce in text first and add phrases about their purpose.

**Answer:** Figure 2 is the schematic representation of our approach and is described and referenced in sections 3.1 and 3.3. HESS guidelines require that figures appear in the main text near the location of their first mention, therefore figure 2 appears in section 3.1. The parts of the figure relating to PCA and GMM are explained in section 3.3, including phrasing of their purpose. The purpose of the PCA and GMM are also stated in the figure caption. If the referee agrees that this is sufficient description of the purpose, we would prefer to keep the figure caption as is.

Equation 2- Is integral or summation better to denote this?

**Answer:** We find the integral better. The integral is physically more correct, whereas the summation of discrete measurements is both practical necessity and convenience.

Ln 175- Which ones are "normalized or standardized"? What is the difference?

**Answer:** Indeed our use of both terms here is confusing. We have edited the text to only use the term "normalised", and also adjusted the phrasing to make it clear that the standardised ranges are part of the definitions of the indices. The sentence now reads: "Here, we use Aich's HI [...] and the simple HI [...], as both have standardised ranges which allows for direct comparison."

Ln 184. "delivered before or after streamflow peak" is interesting! I hope more is discussed.

**Answer:** Thank you! The sediment peak ratio ended up being lumped into the second principal component relating to hysteresis, and therefore has to elude discussion.

Ln 201- "studies" please cite the relevant ones or omit them.

Paragraph 200-206. Some of this paragraph might fit better in the discussion. Authors discretion, but worth consideration (see further comments below).

**Answer:** We propose to shorten and combine the two paragraphs:

"Inter-event effects such as sediment accumulation or exhaustion are important for suspended sediment transport but difficult to quantify. The flow peak ratio $Q_{peak,ratio}$ introduced by Haddadchi and Hicks (2020) is one metric that attempts to quantify inter-event effects:
[Equation]
We modified this metric by using the log-ratio, rather than the ratio, of streamflow peak of the last event $Q^{i-1}_{max}$ and the current
event $Q^{i}_{max}$.

We also attempt to account for inter-event effects with a new metric, the inter-event index (IEI), defined as the ratio of the SSC peak of the previous event $SSC^{i-1}_{max}$ in mg l$^{-1}$ and the time between the end of the last event $t^{i-1}_1$ and the start of the current event $t^{i}_0$ in hours:
[Equation]"

**Comments on results:**

Ln 273- "2"021? Also 22019 needs units

**Answer:** The unit for the number 22019 is metric tonnes. It got misplaced during editing, we have now corrected this.

Ln 282- "Minor data gaps..." I do not understand this sentence.

**Answer:** Thank you for pointing out this confusing phrasing. The sentence now reads: "As PCA can only be applied on complete data, 16 events with minor data gaps had to be discarded, leading us to consider a total of 959 events for the PCA and event clustering." We hope the meaning is now clear.

Ln 284-285- " explains 35%..." do not understand. "...this combination of parameters together explains 35% of the data"

**Answer:** The first principal component explains 35% of the total variance. We have split the sentence in two to make this clearer. The passage now reads:

"We find that PC1 (Fig. 5a) is strongly tied to streamflow and suspended sediment magnitude (i.e. mean and peak event SSC, total suspended sediment mass exported, mean and peak event streamflow, and total streamflow volume). PC1 explains 35 % of the total variance in the data."

Section 4.3-4.4 It seems that these sections are quite closely linked. Should they be combined? Also, while reading section 4.4, I lost track of which cluster was which. Maybe state the characteristics of each cluster in a paragraph. Also, this seems closely linked with some aspects of the discussion (5.1). I always seem to struggle with organizing this sort of information, and linking the results and discussion well, in my papers, but think that this can be streamlined, improving the paper.

**Answer:** Sections 4.3 and 4.4 are related, but keeping these sections separate improves readability for readers searching for specific information in the manuscript. We suggest to change the headings of section 4.3 to "Selection of best cluster model" and 4.4 to "Cluster characteristics", to better represent the contents of the sections. With these headings and as the sections appear directly after one another in the manuscript, the connection between them should be clear to the reader.

Table 4. Related to the last point. It seems that clusters are in fact event types. Why not give them the same wording to clarify? Also, can clusters/types be given a name like "RAIN" or "FREEZE-THAW" to help the reader follow?

**Answer:** We opted for leaving the presentation of the cluster characteristics in the results (Section 4.4.), and then only refer to the clusters as types when we actually interpret them, which we believe is firmly discussion territory (Section 5.1.). This allows readers to develop alternative interpretations of the clusters instead of being confronted with our interpretation only.

The names of each event type is given in the section heading for the interpretation of each event type and again in the summary in Table 5. To aid the reader, we also suggest to add the event type names consistently to all mentions of event type 0, event type 1, etc. in the sections following the event type interpretations.

Example: "Compared to rainfall-melt extremes (type 1), high melt events (type 0) are ..."

Ln 363- "glacier melt... streamflow." How is this known? citation or analysis in the paper.

**Answer:** The phrasing is perhaps confusing here. We intended to make a more general statement which ties in with the first part of the sentence, namely that in the high melt season from July to August, glacier melt contribution to streamflow is at its highest. We propose to change the sentence to make this meaning clearer:

"Given that type 0 events mainly took place during the high melt season from July to August, glacier melt was likely a significant streamflow component."

Ln 375- "channel bank erosion... elevating SSC." These processes seem like possibilities, however, it is unclear to me how they are separated from subglacial sources or land sliding or how the difference between the processes would be known.

**Answer:** We know from field observations that channel erosion occurs at points along the banks, and especially in the proglacial areas. However, we do indeed not know for sure that channel bank

erosion occurs during specific events. With this sentence we intended to indicate that channel bank erosion is a possible process that may contribute sediment during rainfall-melt extremes (type 0). We propose a slight rephrasing of the indicated lines to make this meaning clearer:
"In addition to these processes [i.e. rainfall, snow and glacier melt driven erosion], the high streamflow rates during type 0 events are likely to induce channel bank erosion thus further elevating SSC. Signs of bank erosion has been observed particularly in proglacial areas of the higher catchment reaches, where streams cut through deposits of unconsolidated sediments."

Section 5.1.2- I wonder if these events somehow result from sediment access. Papers such as Vergara et al., 2022, Li et al, 2021, Delaney et al. 2018 and others suggest that increased melt may cause increased sediment transport by accessing sediment stored at high elevations, as opposed to the quantity of water discharge. Therefore, I am also a bit skeptical of the paper's definition of "event" in evaluating these processes. In Vergara et al, 2022, it seems like there is a case of low water discharge, but high sediment concentration. Would these events be considered/captured here?

**Answer:** Events with low water discharge but high sediment concentrations will be detected by our routine. See our reply above to the event definition comment.

Figure 5b- Where does the SWE color bar feature in the figure? Or does this suggest the absence of change in snow at this period? Also, can changes in SWE be confidently assessed of day-long events? Seems like a short timescale to evaluate these changes, is this true?

**Answer:** We assume that the referee means Figure 10  here. For Figures 10 to 13, panel b refers to the change in SWE from the first to the last day of the events. In the case of figure 10, this does indeed suggest an absence of a change in snow during this period. However, we acknowledge that the coarse resolution of the data both in space and time (temporal resolution is 1 day and spatial 1 km), mean that the quantities indicated in the figure are uncertain. As such the change in SWE is only used qualitatively to indicated whether an event has significant snow melt contribution or not as with the examples in Figure 11.

Ln 424-425- This sentence does not make sense to me.

**Answer:** We suggest this phrasing to make the intended meaning clearer:

"What distinguishes type 3 from high melt events (type 1) are lower streamflow and SSY magnitudes, and their timing in the late melt season, with 80% occurring after mid-August."

Sections 5.3, 5.4, 6- I am very happy to see that limitations are discussed, however, these sections seem like a mix of conclusions, limitations, and future work. It seems to me that this could be streamlined and condensed.

**Answer:** Thank you for your suggestion. We propose to remove the fourth paragraph, which in hindsight adds little. That way section 5.3 "Benefits and limitations of methodology" first addresses the advantages of the approach and then the limitations and disadvantages.

Section 5.4 is perhaps more aptly named "Implications of findings for future studies"?

Line 532- "events... the bulk of suspended sediment yield..." I am curious what the characteristics of the non-events periods are... Maybe beyond this paper, but just curious.

**Answer:** This is an interesting idea, but we feel this is beyond the scope of this paper. However, to satisfy the referee's curiosity we attach some figures that show some of the characteristics of the non-event periods.

**Attachment to Authors' reply to Anonymous Referee #1 from 26 Jan 2024**

[Figure]

*Figure 1: Event numbers and annual suspended sediment yield*

[Figure]

*Figure 2: Event type frequency over the study period*

[Figure]

[Figure]

[Figure]

*Figure 3: Difference in time steps during events and outside of events (non-events)*

**Authors' reply to Anonymous Referee #2 from 21 Feb 2024**
https://doi.org/10.5194/hess-2023-300-RC2

**General comments:**

This article aims to derive a typology of sediment-discharge events in the Vent-Rofental basin in the Ötztal Alps, Austria, using a clustering approach. By analyzing continuous and high-resolution records of suspended sediment concentration since 2006, the researchers employed principal component analysis and a Gaussian mixture model to identify and classify different types of events based on their sediment-discharge characteristics. The study's primary objectives were to understand the influence of catchment state, antecedent conditions, and hydrometeorological drivers on event-scale suspended sediment dynamics in the upper Ötztal region. The research questions addressed in the study included identifying key sediment-discharge characteristics that differentiate event types, determining shared hydrometeorological drivers for events of the same type, examining associations between event types and diagnostic antecedent conditions (e.g., dry vs. wet, cool vs. warm), and assessing the contribution of each event type to the annual suspended sediment yield.

The significance of this study lies in providing an objective method to assess the contribution of different drivers to annual sediment yields in mountain rivers. It offers insights into the sediment transport dynamics in mountain rivers, particularly in the context of declining glacial influence. Understanding and managing sediment dynamics in mountain rivers are crucial for effective river management, and this research contributes to that understanding. Additionally, the study demonstrates a reproducible approach to objectively estimate the diversity of event-scale suspended sediment dynamics in mountain rivers, which can serve as a foundation for further research. Overall, I think this is a nice and important work that is suitable for HESS. I would recommend minor revision.

**Major comment 1:**
The Method section should provide more detailed information, particularly regarding the rationale behind the numbers mentioned in the method's implementation. For instance, in Lines 150-155, it would be beneficial to clarify whether a sensitivity analysis was conducted for determining the threshold at the 90th percentile P90 of SSCt. Additionally, in Lines 195-200, the statement "...... peak prominence of 500 mg/l......" should be elaborated upon, as the value of 0.5 g/l appears to be relatively small considering the data presented in the accompanying figure. It would be helpful to explain the specific significance or reasoning behind this particular value. Furthermore, in Line 254, it is important to note that no information was found in the manuscript regarding the utilization of the elbow method for validating the number of clusters.
**Answer:** Thank you. In attempting to be concise, we may have left out important details for the reader to understand the key points of the method. We clarify as follows:
Our motivation for choosing the $90^{th}$ percentile as the SSC-threshold was two-fold. First, as explained in Lines 148-151, the delineation of events essentially splits the entire time series into events at local streamflow minima. Thus the peak SSC-threshold was needed to filter out small suspended sediment pulses, leaving us with events of a sufficient sediment magnitude to be of interest. Second, as clustering is a data-driven approach, we wanted to have a sufficiently large number of events to make the clustering more robust. In our opinion the choice of SSC-threshold at the $90^{th}$ percentile (P90) is a reasonable trade-off between the magnitude criteria and while not being too restrictive and excluding too many events. With the P90 threshold we are left with 38% of the original number of events. At higher thresholds (e.g. $95^{th}$ percentile or $99^{th}$ percentile) the number of events left sharply drops off (see attached Figure 1).
We will add the following sentence for clarification:
"The choice of $P_{90}$ as the peak SSC threshold was a trade-off between ensuring a sufficient number of events for the clustering while keeping only larger events."
The minimal peak prominence is how much higher the peak has to bee compared to preceding and subsequent values in order to be identified as a peak. We selected the minimal peak prominence for streamflow and SSC based on visual inspection of the event hydro- and sedigraphs. A minimal peak prominence of 0.5 g/l is rather conservative, as the median SSC range of events is 1.5 g/l, meaning that the minimal peak prominence is about a third of this range. Thus, out of 976 detected events, only 196 had more than one detected SSC peak.

Our suggested edit to the statement on peak prominence to clarify these points is:
"To calculate the SQPR we set a minimal peak prominence of 500 mg l$^{-1}$ for SSC peaks, which corresponds to about a third of the median SSC range of events, and 2 m$^3$ s$^{-1}$ for streamflow peaks, which corresponds to about an eight of the median streamflow range of events. The minimum distance between peaks was set to one hour. The selection of these parameters for peak detection were based on visual inspection of the hydro- and sedigraphs of the detected events."
Regarding the use of the elbow method, we suggest adding the following sentence to the methods section for clarification:
"The elbow method is a heuristic approach for determining the optimal number of clusters, where the «elbow» or break in the score curve is used to select $K$."

**Major comment 2:**
The structure of the article requires a minor adjustment. Section 5.1 appears to be better suited as a section that describes the results obtained from the three steps: (1) the detection and characterization of events; (2) grouping similar events via clustering; and (3) evaluation and interpretation of these clusters as event types. Placing Section 5.1 after presenting the results of these three steps would enhance the overall logical coherence of the article.
**Answer:** Thank you for this comment. Our opinion is that the detection and characterisation of events (section 4.1), the results of the principal component analysis (section 4.2), the choice of best cluster model (section 4.3) and description of cluster characteristics (section 4.4) firmly belong in the results section. The separation of the cluster charactistics (section 4.4) and interpretation of clusters into event types (section 5.1) was purposely done to keep the distinction between results and discussion. This allows readers to develop alternative interpretations of the clusters instead of being confronted with our interpretation only.
We would argue that the structure proposed by the referee is already the structure of the paper: (1) the detection and characterisation of events are in section 4.1, (2) grouping similar events via clustering (including the transformation of event metrics into principal components) in sections 4.2 and 4.3, and the results of the clustering in sections 4.4, (3) the evaluation and interpretation of clusters as events in section 5.1. The logical order is the same as proposed by the referee.

**Major comment 3:**
To enhance the generalizability of the methods and the readership of the research results, it would be better to compare them with other studies also focused on the cryosphere. For example, in High Mountain Asia, sediment pulses triggered by rainstorm, glacier melt water and snowmelt have also been identified and discussed. Do you think your method can also be applied in High Mountain Asia and other cryospheric regions? This would help attract a wider readership of this paper.
• Zhang, T., Li, D., East, A. E., Kettner, A. J., Best, J., Ni, J., & Lu, X. (2023). Shifted sediment transport regimes by climate change and amplified hydrological variability in cryosphere-fed rivers. Science Advances, 9(45), eadi5019.
• Li, D., Lu, X., Overeem, I., Walling, D. E., Syvitski, J., Kettner, A. J., ... & Zhang, T. (2021). Exceptional increases in fluvial sediment fluxes in a warmer and wetter High Mountain Asia. Science, 374(6567), 599-603.
**Answer:** Thank you for reminding us of the importance of comparing our results with other cold mountain regions. We already have some comparisons with studies from North America in the discussion. We will incorporate the suggested references, as well as some recent work from the Andes, to better reflect the wider scope of warming driven changes in fluvial sediment transport in high mountain areas.
Our method can be applied to any catchment where sufficiently long time series of streamflow and suspended sediment are available, and is not restricted to the cryosphere. However, as we stress in section 5.3, for the interpretation of the event types data representing hydrometeorological forcings, catchment state and antecedent conditions are needed.

**Minor comments:**
Fig. 2: It would be more visually appealing if the font size and labels were increased.
**Answer:** We will increase the font size as much as the figure allows.

Fig. 3: The detection results look impressive, but I have a question regarding why the peaks after 2010-08-13 are not detected as the maximum values.

**Answer:** The two smaller peaks after 2010-08-13 are not detected as events, because their peak SSC is below the threshold. Or to be precise: they were removed based on the SSC-threshold criteria in the event detection (bullet point 3 in Section 3.1).

Fig. 6: It would be valuable to include validation using either the elbow method or silhouette algorithm.
**Answer:** We already used the elbow method when determining the number of clusters (see mention of "elbow point" in first paragraph of section 4.3). We hope our response to Major comment 1 clarifies this. We also propose to add the silhouette score to our validation (see attached figure). We suggest updating the last paragraph of section 4.3 accordingly:
"However, for both the VRC and silhouette score, the spherical-type models consistently outperform all others (Fig. 6b-c), with a clear peak for VRC at K=4, and a minor peak for the silhouette score. The BIC also indicates that four clusters with the spherical covariance type model is reasonable, which is why we selected this variant for our interpretation."

Fig. 7: Add annotations to the violin plot to explain the discrepancy between the mean event cluster shape of "i" and "k" and the actual shape of the clusters.
**Answer:** We propose adding the following sentence to the caption for clarification:
"Where the timing of peak streamflow and SSC is inconsistent within the cluster, the mean event cluster shape is flattened (e; g; i; k)."

Fig. 10, 11, 13: The temperature and rainfall axis scales and labels need to be clearer.
**Answer:** We have added a legend and changed the axis labels and hope the figures are now clearer to read (see attached updated figures).

**Attachment to Authors' reply to Anonymous Referee #2 from 21 Feb 2024**

[Figure]

*Figure 4: Sensitivity of event detection to choice of peak suspended sediment concentration (SSC) threshold.*

[Figure]

Figure 5: Proposed changes to figure 6 in manuscript

[Figure]

Figure 6: Suggested update of figures 10-13.

---

## Author Response (AR2)

**Authors' reply to Anonymous Referee #1 from 04 Jul 2024**

**General comments:**

I thank the authors for addressing the comments presented in the first version.

While I believe that the manuscript has improved, I do not get the sense that the major comments I addressed have been appropriately integrated into the article. Furthermore, there are some additional comments that I wish to bring to light.

Below I present comments related to the text and revisit these topics in light of the author's response.

In addressing these, I believe the paper would be a valuable contribution to the literature.

**Answer:**
Thank you for your continued engagement with our manuscript and for your thoughtful comments and suggestions. We appreciate your acknowledgment of the improvements made in the revised version, and regret that some parts of the revisions did not fully address all of your major comments. We understand the importance of thoroughly integrating your feedback to enhance the manuscript's clarity and impact. We are committed to addressing these concerns more comprehensively in our updated version of the manuscript.

We also appreciate the additional comments you have provided. Below, we address your comments in detail and outline how we plan to incorporate your feedback. Thank you once again for your constructive feedback and for helping us improve our work.

**General scientific comments:**

1. The duration of the time series:

Would it be possible to address processes, such as sediment exhaustion, that could affect the characteristics of events over the time series see Zhang et al. 2022 and Antonniazza and Lane 2021. The authors demonstrate that their method is based upon individual events, which I think is the correct way. Rather, I wonder if the predictor variables could change over the study period given processes not associated with hydro-meteorological conditions, i.e. evolving sediment access. The authors state as much in line 518 "mulitple processes... same signal." Furthermore, event type 4 seems to be access-driven, therefore could this access change through the time series?
Would examining the hysteresis of events through time address this?

I do not think this can be fully addressed in the manuscript, but I would expect this to be discussed.

**Answer:** We have now added the following section to the "Benefits and limitations of methodology" section.

"Sediment dynamics in glaciated catchments are subject to long-term changes in driving processes and sediment availability (Zhang et al., 2022; Antoniazza and Lane, 2021). Annual sSSY and discharge in the Vent-Rofental catchment have remained largely stable during the study period (2006-2021) compared to long-term trends (Schmidt, 2023; Schmidt et al., 2024, 2023), especially

if considering the inter-annual variability. We infer that ongoing long-term changes in the catchment have had only minor effects on our results.

To confirm that the sediment-discharge event characteristics stayed constant over the study period, we checked whether annual averages of event metrics and PCs changed over time and found no significant trends (Mann-Kendall test with 5 % significance level). Likewise, the total annual sSSY and sSSY contributed by rainfall-melt extremes (type 0), freeze-thaw-modulated events (type 2) and late season glacier melt events (type 3) do not show significant trends. However, the annual sSSY contribution of high melt events (type 1) decreased significantly over the study period by $-31$ $t$ $km^{-2}$ $a^{-1}$ (Fig. 14c). This decrease in the annual contribution of high melt events (type 1) is less due to fewer events occurring, but rather a consequence of the average SSY magnitude of type 1 events decreasing. Does this trend influence the detection event types? If the magnitude of high melt events (type 1) had continued to decrease over time, then melt-driven events later in the study period would have differed enough from earlier ones to be separated by the cluster analysis. The advantage lies in considering events individually, which allows our methodology, at least to some extent, to deal with long-term trends."

2. Definition of event types and the ability of the method to capture supply-limited events:

I do not believe that the author's response to this comment was fully adequate and that their justification in the review response was not presented in the manuscript. Some of the responses to my comment in paragraphs 138 to 147 could fit well in an "Experiment limitation" section. However, the response to the general comment did not adequately address the concerns that I presented.

The authors suggest that supply-limited events unrelated to water discharge increases might be rare in this catchment, and these events could be rare in general for that matter. However, this at the very least needs justification and may limit the transferability of their methodology to other catchments. For instance, they state that "our methodology is transferrable... any catchment with subdaily time series" (Ln 478). This seems misleading given the event definition. The methodology might be transferable, however, the event definition should be considered in each case, in my opinion.

Better support for this event definition and its limitations is needed.

**Answer:** We agree that for some catchments the transferability of the methodology is dependent on the event definition and parametrisation detection routine. We have added the following text in the "Benefits and limitations of methodology" section, which we hope makes this clearer:

"Given that fluvial sediment transport is primarily driven by hydrology, it is reasonable to delineate events by streamflow. Single SSC spikes without a corresponding increase in streamflow are expected to be rare, at least in the Vent-Rofental catchment. However, when applying our method to other catchments, other conditions may apply. Consider a mass wasting event without hydro-meteorological drivers, such as an earthquake-triggered landslide. Our event detection would lump the resulting sediment pulse together with an unconnected streamflow pulse. Thus, in catchments where such processes are thought to be relevant, our methodology may be less appropriate. Similarly, in catchment where sediment exhaustion effects are important, or where this is the focus of research, the event filtering by SSC could overlook supply-limited events. In such cases, it may be appropriate to lower $\theta_{SSC,peak}$ to include more events. However, removing the filtering completely would cease to be an event detection, instead splitting the entire time series into event-sized portions, which we advise against. Moreover, the length of the local-minima search window is a parameter to be adapted to the hydrological and sediment response time of the study catchment. In

short, the appropriate choice of parameters for event detection is crucial. The event definition should be adjusted to align with the focus of the study and the hydrological and sediment transport dynamics of a given catchment."

3. Please consider making a section i.e. 5.1.5 about the "non-events." Here, more sediment seems to be exported than in Type 2 and Type 3 events (Figure 14). Knowing more details about this could be useful.

**Answer:** We now added this section on the non-event periods of the time series and how it relates to the proportion of annual SSY contributed by "non-events":

"The periods of the time series without events are concentrated towards the edges of the monitored SSC period, specifically in May-June and September-October. These periods are characterized by markedly lower streamflow rates (90% below 12.3 m3 s−1) and SSC (90% below 588 mg l−1). Therefore, sediment fluxes too are markedly lower during the non-event periods (roughly between 0.01 and 10 t 15min−1) compared to event periods (roughly between 1 and 100 t 15min−1). However, as non-events occupy the largest part of the time series (Fig. 14d), a notable portion of the annual SSY is exported during these non-event periods, especially in years with fewer events, such as 2013 and 2019 (Fig. 14a)."

We discourage any more discussion of the non-event periods in the manuscript for the following reasons:

Firstly, our study focuses on the characterization and analysis of events of a certain magnitude defined by suspended sediment concentration (SSC). The primary objective is to understand the behaviour and drivers of these larger magnitude events within the context of the Vent-Rofental catchment. Adding too much information on "non-events" would diverge from our main research question and dilute the focus of an already long manuscript.

Secondly, "non-events" are per definition not clearly delineated into time chunks like the events are. Rather they are simply the parts of the time series outside marked increases in SSC (i.e. where SSC goes above the threshold of 1196.5 mg/l). Therefore, it is difficult find a meaningful way to compare events and non-events, apart from comparing the overall distributions of the time-steps during and outside of events as we did in the response to the first round of reviews (see https://doi.org/10.5194/hess-2023-300-AC1). This information is included in a simplified manner in the new section added to the manuscript.

We hope this explanation clarifies our rationale for not including a more information on "non-events".

4. The writing is thorough and generally clear. However, there are places where it could be condensed. Please consider.

**Specific scientific comments:**

Ln 75 - reference problem in document.

**Answer:** The problem was due to a preprint that has since been published, we corrected it now.

Ln 167- "studies" please cite the relevant ones or omit them.

**Answer:** We have now included a selection of studies that use duration and magnitude related metrics.

Figure 8. Can the yellow of Cluster 2 be changed to green? this will help it appear in printed versions.

**Answer:** The colours have been carefully selected to be colour-blind friendly while trying to consistently use the same colours to denote the same things throughout the manuscript. Changing the colour of cluster 2 to green would be problematic for colour-blindness.

Paragraph 367- 375: Paragraph can be streamlined? Can the figure or manuscript be referenced to support the first two sentences?

**Answer:** We have adjusted the paragraph to bring up the main point first, and added a reference to Figure 8 to illustrate the timing of the events.

Figure 11. Where is precip? It is a bar plot on top of other figures, but zero here. Mention in the caption or remove axis?

**Answer:** We have added the information to the caption to make it clearer to the reader that there was no precipitation observed, not that there were no observations.

Lines 420-423. Please make the statements of these processes more specific. The connection between "complexity of event shape" and "freezethaw on driving hydrologcial processes" is not clear to me.

**Answer:** We have attempted to make our meaning more clear. The paragraph now reads:

"As with the rainfall-melt extremes (type 0), the complex event shape and high variability in sediment-discharge characteristics within this event type (Fig. 9c) primarily reflect multiple drivers of streamflow and SSC. Unlike rainfall-melt extremes, the complex event shape of type 2 events may also partly be the result of a modulating effect of freeze-thaw on the driving hydrological processes (mechanisms discussed below). The effect of freeze-thaw is thus two-fold, first by affecting the hydrological processes themselves and second by constraining the erodible landscape that contributes sediment to the channels. A similar effect of catchment freeze-thaw state regulating suspended sediment dynamics has been proposed for the Tibetan Plateau (Li et al., 2021b)."

Section 5.1.4. Consider renaming these "sediment access events" or "high elevation melt events." The processes are different from the glacier melt events of T1.

**Answer:** These suggestions with names for type 3 events would not reflect the processes and mechanisms we attribute to these events. "High elevation melt events" would not add more than "glacier melt events", since it is clear that glaciers are located at the higher elevations of the catchment. Also we would like to include the "glacier" here, as "melt events" could imply that there is snow melt involved. "Sediment access events" would go against our argumentation that the lower sediment yields of type 3 events are not driven by a lack of sediment supply or sediment access, but rather by transport limited conditions (i.e. lower streamflow rates).

Ln 477 - I think that the comma after types is a mistake.

**Answer:** Yes, it is. We've removed the comma.

Ln 478- Please see the general comment above for this comment.

Ln 509 - "Hysteresis analysis can be a useful tool" -> for what?

**Answer:** "For exploring the temporal dynamics of SSC and streamflow during events." We've updated the sentence accordingly.

Ln 520 - "although... each event type." This sentence can be clearer. For instance, "although the strength of these characteristics varies with each event type."

**Answer:** Now reads: "…, although the strength of this linkage varies with each event type."